# GenShield: Unified Detection and Artifact Correction for AI-Generated Images

Zhipei Xu [1][*]  Xuanyu Zhang [1][*]  Youmin Xu [2][*]
Qing Huang [1]  Shen Chen [2]  Taiping Yao [2]  Shouhong Ding [2]  Jian Zhang [1][3][✉]

## Abstract

Diffusion-based image synthesis has made AI-generated images (AIGI) increasingly photorealistic, raising urgent concerns about authenticity in applications such as misinformation detection, digital forensics, and content moderation. Despite the substantial advances in AIGI detection, how to correct detected AI-generated images with visible artifacts and restore realistic appearance remains largely underexplored. Moreover, few existing work has established the connection between AIGI detection and artifact correction. To fill this gap, we propose GenShield, a unified autoregressive framework that jointly performs explainable AIGI detection and controllable artifact correction in a closed loop from diagnosis to restoration, revealing a mutually reinforcing relationship between these two tasks. We further introduce a Visual Chain-of-Thought based curriculum learning strategy that enables self-explained, multi-step "diagnose-then-repair" correction with an explicit stopping criterion. A high-quality dataset with large-scale "artifact-restored" pairs is also constructed alongside a unified evaluation pipeline. Extensive experiments on our correction benchmark and mainstream AIGI detection benchmarks demonstrate state-of-the-art performance and strong generalization of our method. The code is available at https://github.com/zhipeixu/GenShield.

This work was supported in part by Shenzhen Science and Technology Program (JCYJ20241202125904007), Guangdong Provincial Key Laboratory of Ultra High Definition Immersive Media Technology (2024B1212010006), Shenzhen Science and Technology Program (SYSPG20241211173440004) and Outstanding Talents Training Fund in Shenzhen. [*]Equal contribution. [✉]Corresponding author. [1]School of Electronic and Computer Engineering, Peking University [2]Tencent Youtu Lab [3]Guangdong Provincial Key Laboratory of Ultra High Definition Immersive Media Technology, Shenzhen Graduate School, Peking University. Correspondence to: Jian Zhang <zhangjian.sz@pku.edu.cn>.

*Proceedings of the 43$^{rd}$ International Conference on Machine Learning*, Seoul, South Korea. PMLR 306, 2026. Copyright 2026 by the author(s).

## 1. Introduction

With the development of deep learning (Li et al., 2025a; 2026a; Zhang et al., 2026a), particularly generative models (Wu et al., 2025b; Labs et al., 2025; Liu et al., 2025; Seedream et al., 2025; Comanici et al., 2025), highly realistic images can now be created with increasing ease and are becoming increasingly difficult to distinguish from real photographs. As AI-generated images (AIGI) rapidly permeate social media, journalism, digital art, and online commerce, concerns regarding authenticity, trust, and visual credibility have become more prominent than ever. The widespread adoption of image generation models has fundamentally altered the visual content ecosystem, where AIGI and camera-captured photographs now coexist at unprecedented scale. In such a setting, the ability to reliably determine whether an image is generated by an image generative model or captured by a real camera is no longer merely a technical challenge, but a critical requirement for a wide range of real-world applications, including misinformation detection, digital forensics, and content moderation.

AI-generated image detection (Yan et al., 2024; Zhou et al., 2025; Wen et al., 2025) aims to determine whether an image has been produced by an image generator, typically by leveraging statistical cues, pixel-level artifacts, and semantic-level anomalies. A related and more fine-grained problem is synthetic image artifact correction (Wang et al., 2025; Fang et al., 2024), which focuses on identifying and localizing non-natural regions and then correcting them to restore a more realistic appearance, such as structural inconsistencies, violations of physical laws, and local distortions. However, most existing studies (Kang et al., 2025; Fang et al., 2024; Wang et al., 2025; Shao et al., 2025) primarily emphasize more accurate detection and localization, and commonly follow a pipeline that highlights artifacts with a bounding box or mask and applies a frozen inpainting model for local repainting. This design has several limitations: (i) the correction quality heavily depends on precise localization, which can be unreliable in practice; (ii) a frozen inpainting model becomes a performance bottleneck and limits the upper bound of correction; and (iii) inpainting often produces seams or inconsistencies between the repainted region and surrounding context, potentially introducing new artifacts. As a result, mask-free, end-to-end artifact cor-

rection remains largely underexplored. Moreover, existing datasets (Kang et al., 2025) are largely detection-oriented: they provide artifact localization and textual descriptions, but rarely include paired restoration targets, which limits the progress of artifact correction methods.

Most existing studies overlook the potential synergy between correction and detection, even though **these two processes can naturally reinforce each other when modeled jointly**. On the one hand, incorporating detection enables accurate anomaly identification and fine-grained artifact localization, which provides essential guidance for correction. This guidance steers the model to focus on truly problematic regions and remove non-natural cues without unnecessary changes. On the other hand, incorporating correction introduces a strong generative prior, strengthening the model's ability to reconstruct realistic images and thereby sharpening its sensitivity to subtle artifacts that are otherwise hard to distinguish. Therefore, it is natural to frame the task as a joint anomaly detection and repair problem, where the model not only detects and localizes artifacts but also corrects them to restore visual authenticity. Crucially, this setting reveals a mutually reinforcing relationship between understanding and generation, as diagnostic understanding guides targeted correction, and generation-based correction improves sensitivity for detection.

In recent years, the field of unified multimodal understanding and generation has witnessed rapid development. For example, Emu3 (Wang et al., 2024) tokenizes images, text, and videos into a shared discrete space and trains a single transformer via next-token prediction. Show-o (Xie et al., 2024) unifies autoregressive and discrete diffusion modeling within one transformer for mixed-modality tasks such as VQA and text-to-image generation. BAGEL (Deng et al., 2025) presents a mixture-of-transformer-experts (MoT) architecture that harmonizes autoregressive language models with rectified flow, utilizing a dual-encoder design to effectively balance high-level semantic understanding with high-fidelity image generation. These unified architectures suggest that understanding and generation can boost each other. In this context, detection and artifact correction exhibit a similar understanding–generation duality, providing strong support for jointly modeling them within a unified framework.

Based on the above analysis, we propose GenShield, a unified autoregressive framework built upon a MoT architecture. GenShield consists of two specialized experts for AI-generated image detection and artifact correction, operating on a shared multimodal backbone with shared self-attention. Within this unified architecture, GenShield supports two tightly coupled tasks: (i) structured AIGI detection with explanatory rationales, and (ii) artifact correction, which can be performed either via instruction-guided edit-

ing or through an iterative visual chain-of-thought (VCoT) diagnose–then–correct process that alternates between diagnostic reasoning and targeted image refinement. To effectively train the model, we design a two-stage curriculum learning strategy. Stage 1 focuses on instruction-guided correction to establish stable generative priors, while Stage 2 introduces multi-step VCoT self-correction. Throughout both stages, the AIGI detection task remains active, enabling joint optimization of detection and correction. Our contributions are summarized as follows.

❏ (1) We propose GenShield, a unified autoregressive framework that connects AI image detection and artifact correction. By coupling semantic understanding with pixel-level reconstruction, our method forms an end-to-end loop from artifact diagnosis to authenticity restoration, showing the synergistic effects between detection and correction.

❏ (2) We design a VCoT-based curriculum learning strategy. By transitioning from instruction-guided correction to multi-step self-correction with an explicit stopping criterion, while keeping AIGI detection active throughout training, we establish a "diagnose–then–correct" paradigm that reduces learning complexity and improves logical transparency.

❏ (3) We construct a specialized, high-quality dataset GenShield-Set tailored for unitied AIGI detection and correction. By leveraging explicit defect descriptions to guide advanced editors, we generate large-scale and precisely aligned "artifact-restored" image pairs. This fills a critical data gap and enables models to learn the mapping from synthetic anomalies to realistic images.

❏ (4) Experimental results demonstrate that our GenShield achieves state-of-the-art performance on AIGI detection tasks, while its artifact correction capability surpasses that of advanced closed-source generators.

## 2. Related Works

### 2.1. Synthetic Image Detection

Early synthetic image detection methods (Wang et al., 2020; Zhong et al., 2023; Frank et al., 2020; Tan et al., 2023; Wang et al., 2023; He et al., 2026b; 2024; 2026a; Zhou et al., 2023) mainly use methods such as frequency domain artifacts, reconstruction residuals, and local pixel correlation to mine the low-level forgery clues introduced in the generation process. With the rise of vision foundation models (Radford et al., 2021; Zhang et al., 2022), semantic-level cues are increasingly adopted for detection. CLIP-based methods such as UniFD (Ojha et al., 2023), C2P-CLIP (Tan et al., 2025b) and FatFormer (Liu et al., 2024a) exploit image-text contrastive features to improve generalization, while self-supervised models like DINO offer transferable visual representations without relying on text supervision (Guillaro

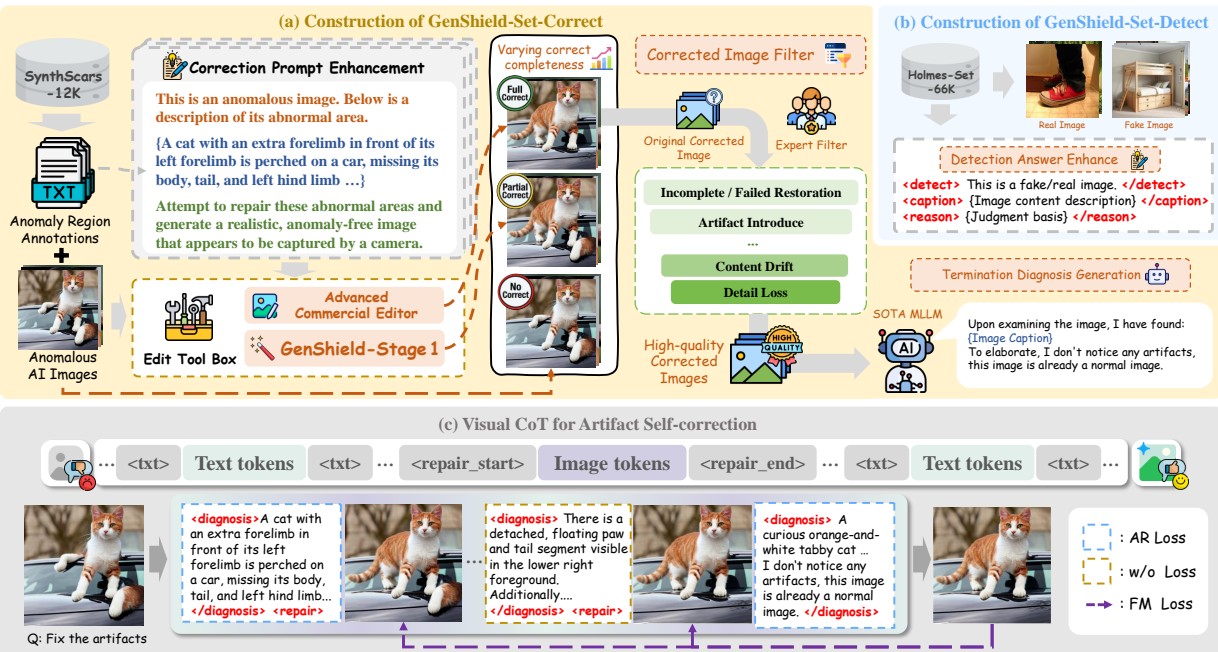

*Figure 1.* **Overview of GenShield-Set Construction.** (a) Correction data generation with prompt enhancement, expert filtering and termination answer generation. (b) Detection data enhancement with structured answers. (c) Interleaved text–image VCoT sequences for iterative diagnosis and correction.

et al., 2025; Cai et al., 2026). With the development of large language models (LLMs) (Bai et al., 2025; Li et al., 2024; Cao et al., 2026; Cai et al., 2025), recent image forgery detection methods have increasingly incorporated LLMs to address the limitations of traditional black-box classifiers in generalization and explainability (Xu et al., 2025a; Liu et al., 2024b; Xu et al., 2025b; Lin et al., 2025b;c). By projecting image content into the language space, these methods generate reasoning-rich textual representations that enhance sensitivity to semantic anomalies. However, most existing methods still treat AIGI detection as an isolated task, optimizing accuracy, robustness, or explanations in separation. In contrast, the synergy between artifact correction and detection remains underexplored. Jointly modeling this complementarity is a promising yet insufficiently studied direction.

### 2.2. Synthetic Image Artifact Localization and Correction

Beyond global real/fake classification, recent studies have begun to investigate fine-grained anomaly understanding in synthetic images by explicitly characterizing and localizing non-natural regions. Zhang et al. (Zhang et al., 2023) study perceptual artifact localization for image synthesis tasks, showing that generation artifacts can be spatially identified rather than only judged at the image level. Building on the need for fine-grained artifact understanding, SynArtifact (Cao et al., 2024) provides a taxonomy of 13 defect types and leverages vision–language models to both classify

and localize these artifacts, enabling more structured diagnosis of generation-induced defects. Moving beyond predefined artifact categories, AnomAgent (Tan et al., 2025a) further targets semantic anomalies and proposes a multi-agent reasoning framework to identify and explain violations of physical laws or commonsense, emphasizing interpretable, human-like forensic analysis. In parallel to these general artifact understanding approaches, HumanRefiner (Fang et al., 2024) narrows the scope to synthesized humans and focuses on biological plausibility, using skeleton pose priors to detect anatomical deformities, and then relies on a frozen generator to perform regeneration or inpainting based on the detected results. While these efforts advance artifact categorization, localization, and explanation, they largely emphasize anomaly diagnosis rather than end-to-end correction, leaving the joint modeling of detection-driven diagnosis and controllable repair an open direction.

### 2.3. Unified Understanding and Generation

Compared with conventional pipelines that treat image understanding and generation as separate processes, recent work (Wu et al., 2025a; Yang et al., 2026a) increasingly unifies them within a single framework, where joint optimization improves overall capability and generalization. Inspired by language modeling, some studies (Chen et al., 2025c; Wang et al., 2024; Yang et al., 2026b) encode images into discrete tokens and generate them autoregressively via next-token prediction. Other approaches (Xie et al., 2024; Zhou et al., 2024) incorporate diffusion mechanisms, using text-

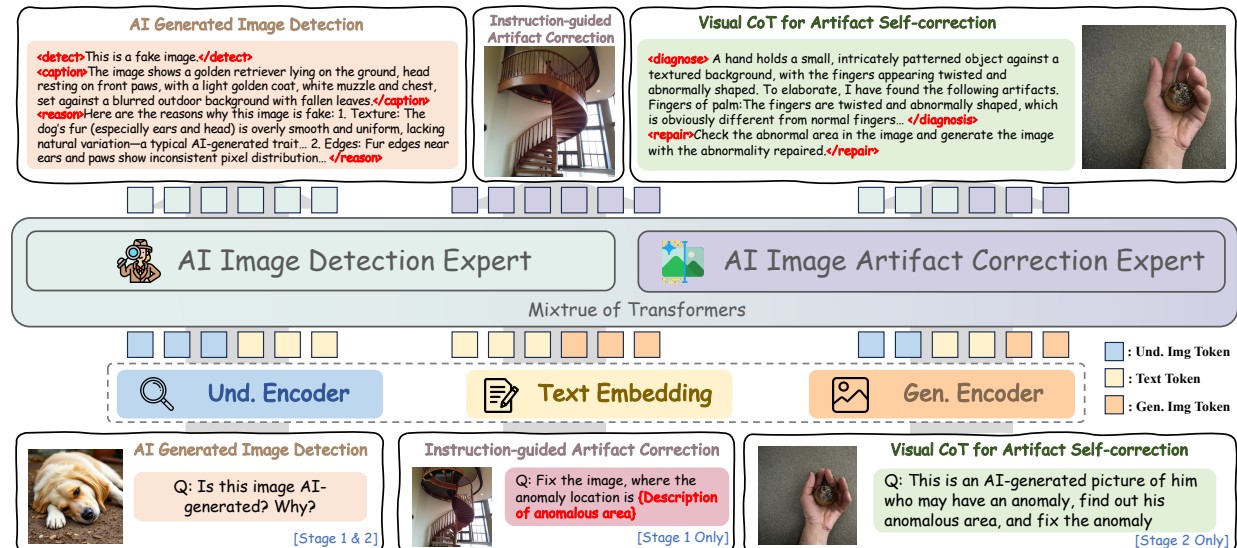

*Figure 2.* **Pipeline of GenShield.** Stage 1 performs AI-generated image detection and instruction-guided artifact correction, while Stage 2 extends correction to an iterative VCoT setting, alternating between multi-step diagnosis and refinement.

token generation as a bridge to model continuous images and improve generation quality. Another line of work (Lin et al., 2025a; Chen et al., 2025a) emphasizes modular designs, decoupling perception and generation to better optimize each component. The recent BAGEL model (Deng et al., 2025) adopts a Mixture-of-Transformers (MoT) design, separating parameters for understanding and generation while sharing self-attention, further boosting performance. This surge of unified understanding-and-generation models makes the synergy between AIGI detection and artifact correction both feasible and compelling.

## 3. Methodology

### 3.1. Construction of GenShield-Set

As shown in Fig. 1, we construct GenShield-Set for joint training of explainable AIGI detection and artifact correction, comprising GenShield-Set-Correct with paired correction targets and GenShield-Set-Detect with structured detection answers.

**GenShield-Set-Correct.** As illustrated in Fig. 1(a), we build GenShield-Set-Correct on top of SynthScars (Kang et al., 2025), which provides anomalous synthetic images together with textual anomaly-region annotations. For each anomalous image $\mathbf{I}_{artifact}$ and its annotation $\mathbf{T}_{diag}$, we first perform correction prompt enhancement, rewriting the annotation into a standardized instruction . We employ an advanced proprietary image editing model to generate corrected image candidates. Since automatic editing may result in under-repair or over-editing, we apply a human expert-based corrected image filter to select high-quality restorations $\mathbf{I}_{correct}$. This filter removes candidates with common failure modes, including incomplete or failed restoration, newly introduced artifacts, content drift, and excessive de-

tail loss, ensuring that the retained repaired images are both artifact-reduced and semantically consistent with the originals. Subsequently, we additionally create a termination diagnosis $\mathbf{T}_{stop}$ for $\mathbf{I}_{correct}$ by prompting Qwen-2.5-VL (Bai et al., 2025) to explicitly state that no artifacts are observed and the image already appears normal. This termination diagnosis is later used in our iterative VCoT process to determine when further correction is unnecessary. Notably, to guide multi-round iterative correction, we use the intermediate results of GenShield (in training stage 1) to generate a set of medium-quality corrected images $\mathbf{I}_{mid}^{(i)}$. These predictions typically apply relatively mild edits and therefore only partially remove the artifacts in the original images. Finally, we obtain over 10K high-quality, precisely aligned tuples $(\mathbf{I}_{artifact}, \mathbf{T}_{diag}, \mathbf{I}_{mid}^{(1)}, \ldots, \mathbf{I}_{mid}^{(N)}, \mathbf{I}_{correct}, \mathbf{T}_{stop})$ as training data. More details are provided in the ***Appendix D***.

**GenShield-Set-Detect.** As illustrated in Fig. 1(b), to train the detection expert, we construct GenShield-Set-Detect primarily from Holmes-Set (Zhou et al., 2025), which contains real and AI-generated image with detailed explanatory detection annotations. We further perform detection answer enhancement by converting the raw annotations into a unified, structured format that includes an authenticity prediction (<detect>), an image caption (<caption>), and an evidence-based justification (<reason>). Finally, we obtain 66K image–text pairs for training.

### 3.2. Overview of GenShield

**Motivation.** Nowadays, with the growing synergy between multimodal understanding and image generation (Deng et al., 2025; Xie et al., 2024), we explore whether AI image detection can be jointly improved with a complementary generative task. We observe that artifact correction

is highly aligned with AIGI detection: detection produces structured and fine-grained diagnostic cues that can guide targeted correction, while correction encourages the model to reconstruct realistic distributions, thereby improving its sensitivity and generalization to subtle artifacts. However, existing pipelines (Fang et al., 2024; Wang et al., 2025; Zhou et al., 2025; Kang et al., 2025) typically decouple these two processes or only focus on a single task, which limits controllability and hinders mutual reinforcement.

**Task Definition.** Motivated by the complementary nature of explainable detection and artifact correction, we introduce GenShield, a unified autoregressive framework that connects AIGI detection with controllable restoration in a single model, as illustrated in Fig. 2. Given an input image and a task instruction, GenShield supports two tightly coupled tasks: (i) *AIGI detection*, where the model autoregressively generates a structured decision together with forensic rationales; and (ii) *artifact correction*, which can be carried out either via instruction-guided editing under explicit artifact descriptions or through an iterative VCoT diagnose–then–correct process that alternates between diagnostic reasoning and targeted image refinement with an explicit stopping criterion.

**Model Architecture.** GenShield adopts a mixture-of-transformers (Deng et al., 2025) design with two specialized experts, an AI Image Detection Expert and an AI Image Artifact Correction Expert, built on top of a shared decoder-style multimodal backbone. Both experts operate on the same multimodal token sequence and interact through shared multi-modal self-attention at every layer, which serves as the key mechanism for mutual reinforcement. On one hand, the correction expert injects camera-consistent generative priors into the shared backbone, aligning detection features with real-image statistics and sharpening sensitivity to subtle artifacts. On the other hand, during correction, the detection expert provides repair-oriented diagnostic descriptions that are propagated through the same attention pathway as explicit guidance, enabling more precise edits with better semantic preservation. Together, shared self-attention enables bidirectional information flow between understanding and generation, forming the architectural basis of GenShield's closed-loop synergy between detection and correction.

### 3.3. VCoT-based Curriculum Learning

Synthetic artifact correction must satisfy two competing requirements: accurately identifying and removing non-natural artifacts, while preserving identity, semantics, and global structure. Directly learning an end-to-end mapping from "anomalous image → refined image" often suffers from sparse supervision, uncontrolled edit scope, and semantic drift. To address this, we propose a VCoT-based curriculum learning strategy, which explicitly decomposes

correction into a two-step reasoning pipeline, Diagnose-then-Repair, and progressively builds reliable diagnosis and controllable repair via an easy-to-hard training schedule. In addition, we keep the AIGI detection task active throughout training, allowing detection and correction to mutually reinforce each other.

**Detection Part.** We formulate AIGI detection as autoregressive structured-text generation. Given the image prompt context $\mathbf{x}_{\text{cond}} = [\mathbf{I}_{\text{det}}, \mathbf{T}_{\text{det}}]$ with length $l_{\text{con}}$, the model generates the detection answer as text tokens appended to the context. We train the detection expert by maximum likelihood with an autoregressive objective:

$$\mathcal{L}_{AR}(\theta) = -\mathbb{E}_{(\mathbf{x}_{\text{cond}}, \mathbf{x}_{\text{text}})} \left[ \sum_{i=l_{\text{con}}}^{l-1} \log P_\theta(\mathbf{x}_{i+1}|\mathbf{x}_{1:i}) \right]. \quad (1)$$

Here $\mathbf{x}_{\text{text}}$ denotes the output text, here is the detection results and explanation. $l$ is the total sequence length. The prefix $\mathbf{x}_{1:i}$ contains all previous multimodal tokens. This objective supervises the model to generate structured detection outputs that include both an authenticity decision and evidence-based rationales.

**Correction Part.** We decompose the optimization of the correction component into an easy-to-hard curriculum with two stages: instruction-guided correction and Visual CoT for self-correction.

† denotes the setting following (Zhou et al., 2025), where the artifact features and semantic features are directly concatenated.

*Stage 1: Instruction-guided correction.* In Stage 1, we train the correction expert with strong supervision using instruction-guided correction, which provides a stable learning signal before introducing iterative self-correction. Concretely, the input consists of the anomalous image $\mathbf{I}_{\text{artifact}}$ and a correction instruction derived from the diagnostic annotation $\mathbf{T}_{\text{diag}}$ (a detailed and structured artifact description), and the model learns the mapping $\mathbf{x}_{\text{cond}} = [\mathbf{I}_{\text{artifact}}, \mathbf{T}_{\text{diag}}] \rightarrow \mathbf{I}_{\text{correct}}$. The model is optimized to generate the target corrected image $\mathbf{I}_{\text{correct}}$ in a flow matching manner.

$$\mathcal{L}_{FM}(\theta) = \mathbb{E}_{\mathbf{z}_0 \sim \mathcal{N}(0,\mathbf{I})} \left[ \|v_\theta(\mathbf{z}_t, t|\mathbf{x}_{\text{cond}}) - (\mathbf{I}_{\text{correct}} - \mathbf{z}_0)\|^2 \right], \quad (2)$$

where $\mathbf{z}_t = t\mathbf{I}_{\text{correct}} + (1-t)\mathbf{z}_0$, $v_\theta$ denotes the velocity neural network used for artifact correction with parameter $\theta$, and $t$ is the sampled diffusion timestep. This stage encourages the model to learn a realistic correction prior and to perform localized, semantically preserving edits under explicit guidance, thereby reducing uncontrolled edit scope and semantic drift. The trained model also serves as the initialization for Stage 2, and is used to produce intermediate corrections $\mathbf{I}_{\text{mid}}^{(n)}$ that diversify the starting states for multi-round VCoT training.

*Stage 2: Visual CoT for self-correction.* As shown in Fig. 1 (c), Stage 2 upgrades correction from externally

*Table 1.* Evaluation of AI synthetic image detection ability on Holmes-Set (Zhou et al., 2025). **Bold** indicates the best result, and underlined denotes the second best. [† : the setting following (Zhou et al., 2025), where the artifact features and semantic features are directly concatenated.]

| Method | Janus | | Janus-Pro-1B | | Janus-Pro-7B | | Show-o | | LlamaGen | | Infinity | | VAR | | PixArt-XL | | SD3.5-Large | | FLUX | | Mean | |
|---|---|---|---|---|---|---|---|---|---|---|---|---|---|---|---|---|---|---|---|---|---|---|
| | Acc. | A.P. | Acc. | A.P. | Acc. | A.P. | Acc. | A.P. | Acc. | A.P. | Acc. | A.P. | Acc. | A.P. | Acc. | A.P. | Acc. | A.P. | Acc. | A.P. | Acc. | A.P. |
| *Non-LLM Based AI Image Detector* | | | | | | | | | | | | | | | | | | | | | | |
| CNNSpot | 70.0 | 86.0 | 70.9 | 85.8 | 85.0 | 93.6 | 72.2 | 86.0 | 61.9 | 71.4 | 86.8 | 94.6 | 59.9 | 75.0 | 78.2 | 90.1 | 63.8 | 81.1 | 79.9 | 92.0 | 72.9 | 85.6 |
| AntifakePrompt | 72.2 | 87.4 | 84.3 | 94.0 | 84.8 | 93.1 | 86.2 | 95.5 | 96.2 | 99.4 | 90.7 | 95.6 | 81.7 | 92.8 | 92.8 | 97.8 | 66.1 | 80.8 | 84.0 | 94.6 | 83.9 | 93.1 |
| UniFD | 87.6 | 97.8 | 96.9 | 99.5 | 96.4 | 99.5 | 85.9 | 97.4 | 93.1 | 98.6 | 79.2 | 96.2 | 64.3 | 85.9 | 75.7 | 94.4 | 87.8 | 97.8 | 69.6 | 91.4 | 83.6 | 95.9 |
| NPR | 51.2 | 55.9 | 69.5 | 75.1 | 73.9 | 77.9 | 93.7 | 99.6 | 93.5 | 99.4 | 93.8 | 99.9 | 85.9 | 91.2 | 93.4 | 99.1 | 91.6 | 97.7 | 93.6 | 99.5 | 84.0 | 89.5 |
| LaRE | 70.8 | 99.3 | 74.7 | 97.5 | 95.6 | 99.7 | 80.0 | 99.0 | 91.6 | 99.6 | 77.9 | 99.6 | 98.8 | 100.0 | 82.2 | 99.7 | 94.1 | 99.5 | 84.3 | 99.0 | 85.0 | 99.3 |
| RINE | 89.9 | 98.3 | 98.7 | 99.9 | 97.2 | 99.6 | 98.8 | 99.9 | 99.1 | 100.0 | 99.2 | 99.9 | 85.0 | 97.9 | 98.9 | 99.8 | 97.8 | 99.7 | 97.1 | 99.7 | 96.2 | 99.5 |
| AIDE | 91.2 | 99.1 | 98.9 | 99.9 | 97.8 | 99.8 | 98.0 | 99.8 | **99.4** | **100.0** | 98.7 | 99.9 | 93.6 | 99.3 | 98.6 | 99.9 | **99.4** | **100.0** | 94.4 | 99.5 | 97.0 | 99.7 |
| *LLM Based AI Image Detector* | | | | | | | | | | | | | | | | | | | | | | |
| AIGI-Holmes† | 95.7 | 99.8 | 99.1 | **100.0** | 93.4 | 99.6 | 97.5 | 99.9 | 98.0 | 99.9 | **99.6** | **100.0** | 99.2 | **100.0** | 95.2 | 99.8 | 98.8 | 99.9 | 79.5 | 96.5 | 95.6 | 99.5 |
| FakeVLM | 86.8 | 93.0 | 83.3 | 83.5 | 68.0 | 72.6 | 78.3 | 81.9 | 86.8 | 87.3 | 83.4 | 85.3 | 82.0 | 84.1 | 77.1 | 82.3 | 82.5 | 84.2 | 83.7 | 89.5 | 81.2 | 84.4 |
| Qwen2.5-VL-7B | **99.6** | **99.9** | **99.8** | 99.8 | 63.0 | 77.5 | 84.9 | 90.7 | 80.7 | 89.9 | 98.1 | 98.7 | **99.8** | 99.9 | **99.6** | 99.8 | 98.2 | 98.9 | 54.2 | 66.2 | 87.8 | 92.1 |
| **Ours** | 98.8 | 99.7 | 99.5 | 99.8 | **99.4** | **99.9** | 99.2 | 99.9 | 98.8 | 99.5 | 98.9 | 99.7 | 99.4 | 99.9 | 99.3 | **99.9** | 96.7 | 99.8 | **97.9** | 99.7 | **98.8** | **99.8** |

guided editing to self-correction with multi-step Visual CoT. Starting from a simple prompt $\mathbf{Q}$ (e.g. "Please repair this image.") and an input image $\mathbf{I}_{\text{artifact}}$, the model first generates a repair diagnosis $\hat{\mathbf{T}}_{\text{diag}}$ that describes remaining suspicious regions and artifact cues, and then performs a correction step conditioned on this diagnosis to produce an updated image. The updated image is re-fed into the model to trigger the next diagnose–correct iteration, forming an alternating multi-round process that progressively removes residual artifacts. The process can be formulated as.

$$\underbrace{[\mathbf{Q}, \mathbf{I}_{\text{artifact}}]}_{\text{Initial state}} \rightarrow \underbrace{[\mathbf{T}_{\text{mid}}^{(1)}, \mathbf{I}_{\text{mid}}^{(1)}] \rightarrow \cdots \rightarrow [\mathbf{T}_{\text{mid}}^{(N)}, \mathbf{I}_{\text{mid}}^{(N)}]}_{\text{Intermediate state}} \rightarrow \underbrace{[\mathbf{T}_{\text{stop}}, \mathbf{I}_{\text{correct}}]}_{\text{Termination state}}$$

*Initial state.* We adopt an interleaved image—text training, where a simple prompt is progressively refined into a detailed restoration instruction and simultaneously performing an initial correction of the image, namely $\mathbf{x}_{\text{cond}} = [\mathbf{Q}, \mathbf{I}_{\text{artifact}}] \rightarrow [\mathbf{T}_{\text{diag}}, \mathbf{I}_{\text{correct}}]$. The loss function is $\mathcal{L} = \mathcal{L}_{FM} + \lambda \mathcal{L}_{AR}, \lambda = 0.25$.

*Intermediate state.* To make this self-correction chain trainable, we supervise each produced image of the intermediate state toward the same target $\mathbf{I}_{\text{correct}}$ via the loss function $\mathcal{L} = \mathcal{L}_{FM}$, namely $\mathbf{x}_{\text{cond}} = [\mathbf{Q}, \mathbf{I}_{\text{mid}}^{(i)}] \rightarrow \mathbf{I}_{\text{correct}}$. To be noted, we leave intermediate diagnostic texts $\mathbf{T}_{\text{mid}}^{(i+1)}$ unconstrained in later rounds to encourage free-form yet actionable reasoning.

*Terminate state.* To explicitly instruct the model to determine when to stop restoration, the ground-truth $\mathbf{I}_{\text{correct}}$ is provided as the image input, such that the model preserves the image while generating the termination text $\mathbf{T}_{\text{stop}}$. The loss function $\mathcal{L} = \mathcal{L}_{FM} + \lambda \mathcal{L}_{AR}$, with $\mathbf{x}_{\text{cond}} = [\mathbf{Q}, \mathbf{I}_{\text{correct}}] \rightarrow [\mathbf{T}_{\text{stop}}, \mathbf{I}_{\text{correct}}]$. During inference, when the model is fed with satisfactory results, our GenShield will output the text like " I do not notice any artifacts, this image is already a normal image" and terminate the multi-step restoration process. This design enables the model to automatically decide when further correction is unnecessary, leading to stable multi-step refinement without over-editing.

## 4. Experiment

### 4.1. Experiment Setup

**Implementation details.** We follow a unified multi-task training pipeline with a curriculum schedule. In Stage 1, we jointly train detection and instruction-guided repair. In Stage 2, we keep detection training unchanged while upgrading repair to the Visual CoT diagnose-then-repair formulation. We build our method on top of BAGEL (Deng et al., 2025), which employs a ViT-style encoder (Dosovitskiy, 2020) for visual understanding and whose VAE is from FLUX (Labs et al., 2025). We use AdamW in both stages with a fixed learning rate of $2 \times 10^{-5}$ and a 500-step warmup. We freeze the Und. Encoder and Gen. Encoder, while keeping all other components trainable. More details are provided in the *Appendix E*.

### 4.2. AIGI Detection Performance Evaluation

We evaluate our method on the Holmes-Set (Zhou et al., 2025), which consists of AI-generated and real images across various generative models. We categorize the compared models into LLM-based and non-LLM-based AI image detectors based on whether their architecture incorporates a large language model (LLM). For fair comparison, all methods were retrained on this dataset under identical conditions. Among them, all non-LLM-based detector results are sourced from (Zhou et al., 2025), while we retrain all LLM-based detectors under the same training settings. We use Acc. (accuracy) and A.P. (area under the precision-recall curve) as the detection evaluation metrics.

As shown in Table 1, our method outperforms existing SOTA methods in both Acc. and A.P. across multiple generative models. Specially on Janus-Pro-7B, our approach achieves 99.4% accuracy and 99.9% A.P., far exceeding the second-

*Table 2.* Evaluation of AI-generated artifact correction ability on SynthScars (Kang et al., 2025). **Bold** and underline denote the best and second best result. [\*: our single-step correction variant, $\triangle$: LEGION equipped with an external SDXL inpainting module.]

| Method | GPT-assisted Evaluation | | | | Human Evaluation | | | | Objective Metrics | | |
|---|---|---|---|---|---|---|---|---|---|---|---|
| | Structure ↓ | Physics ↓ | Distortion ↓ | Mean ↓ | Structure ↓ | Physics ↓ | Distortion ↓ | Mean ↓ | HPSv3 ↑ | CLIP-Score ↑ | PickScore ↑ |
| *Closed-Sourced Image Edit Model* | | | | | | | | | | | |
| GPT-Image | 0.24 | 0.13 | 0.34 | 0.24 | 0.28 | 0.20 | 0.20 | 0.23 | 6.09 | 21.83 | 18.70 |
| Nano-Banana-Pro | 0.22 | 0.21 | 0.29 | 0.24 | 0.21 | 0.28 | 0.26 | 0.25 | 5.92 | 21.89 | 18.71 |
| Nano-Banana | 0.25 | 0.24 | 0.32 | 0.27 | 0.20 | 0.18 | 0.27 | 0.21 | 5.77 | 21.40 | 18.64 |
| Seedream | 0.33 | 0.31 | 0.39 | 0.34 | 0.31 | 0.28 | 0.28 | 0.29 | 6.12 | 21.76 | 18.62 |
| FLUX-Pro | 0.27 | 0.33 | 0.29 | 0.30 | 0.21 | 0.29 | 0.20 | 0.23 | 5.98 | 22.07 | 18.68 |
| *Open-Sourced Image Edit Model* | | | | | | | | | | | |
| BAGEL | 0.70 | 0.69 | 0.84 | 0.74 | 0.72 | 0.71 | 0.79 | 0.74 | 4.68 | 21.44 | 18.54 |
| Qwen-Image-Edit | 0.49 | 0.48 | 0.61 | 0.53 | 0.37 | 0.35 | 0.51 | 0.41 | 5.99 | 21.41 | 18.54 |
| Step1X-Edit | 0.43 | 0.37 | 0.48 | 0.43 | 0.39 | 0.36 | 0.41 | 0.39 | 5.47 | 21.33 | 18.60 |
| *Artifact Correction Model* | | | | | | | | | | | |
| LEGION$^\triangle$ | 0.35 | 0.35 | 0.47 | 0.39 | 0.39 | 0.41 | 0.3 | 0.37 | 5.01 | 21.46 | 18.61 |
| Ours* | 0.18 | 0.21 | 0.30 | 0.23 | 0.17 | 0.25 | **0.16** | 0.19 | **6.23** | 22.11 | 18.83 |
| **Ours** | **0.13** | **0.10** | **0.21** | **0.15** | **0.15** | **0.12** | 0.19 | **0.16** | 6.20 | **22.12** | **18.86** |

*Table 3.* Ablation studies on different training strategies. [Det.: Detection; Corr.: Correction]

| Method | Detection | | Correction | | |
|---|---|---|---|---|---|
| | Acc. | A.P. | HPSv3 ↑ | CLIP-Score ↑ | PickScore ↑ |
| Only Det. | 96.4 | 98.1 | - | - | - |
| Only Corr. | - | - | 5.39 | 21.47 | 18.61 |
| Only Stage 1 | 97.1 | 98.8 | 5.93 | 21.91 | 18.81 |
| Only Stage 2 | 97.6 | 99.3 | 5.97 | 21.96 | 18.79 |
| w/o VCoT | 98.6 | 99.7 | 5.99 | 22.02 | 18.82 |
| **Ours** | 98.8 | 99.8 | 6.20 | 22.12 | 18.86 |

*Table 4.* Robustness study of AIGI detection on JPEG compression, Gaussian blur and Resize of GenShield.

| Method | Original | JPEG Compression | | Gaussian Blur | | Resize |
|---|---|---|---|---|---|---|
| | | QF=75 | QF=70 | $\sigma = 1.0$ | $\sigma = 2.0$ | ×0.5 |
| CNNSpot | 72.9 | 63.5 | 62.4 | 64.5 | 61.7 | 59.9 |
| NPR | 84.0 | 52.2 | 51.6 | 56.8 | 53.4 | 74.3 |
| UniFD | 83.6 | 84.7 | 84.0 | 81.0 | 74.9 | 86.3 |
| LaRE | 85.0 | 62.0 | 63.0 | 54.3 | 54.2 | 51.1 |
| AntifakePrompt | 83.9 | 80.1 | 79.7 | 78.2 | 77.6 | 74.5 |
| AIDE | 97.0 | 92.8 | 92.3 | 91.9 | 90.7 | 89.2 |
| RINE | 96.2 | 92.4 | 91.1 | 94.2 | 92.8 | 92.3 |
| AIGI-Holmes | 95.6 | 93.1 | 92.5 | 92.3 | 90.8 | 90.2 |
| **Ours** | **98.8** | **98.0** | **97.8** | **97.5** | **96.9** | **96.1** |

best result of 97.8% accuracy and 99.8% A.P. achieved by AIDE. Our model also leads on the mean performance across all generators, with a 98.8% accuracy and 99.8% A.P.. The results demonstrate the strong performance of our method, which can be attributed to the synergistic integration of explainable detection and artifact correction within a unified framework. By leveraging both detailed diagnostic reasoning and robust generative priors, our approach effectively detects subtle anomalies, leading to its superior performance on the Holmes-Set benchmark. Additionally, an interpretable subjective result, as shown in Fig. 3(c), demonstrates how our model accurately detects that the image is fake and provides a detailed analysis of why it is considered fake from texture, shadows, and lighting. More experimental results can be found in the ***Appendix B***.

## 4.3. Artifact Correction Performance Evaluation

To evaluate the artifact correction performance of our method, we evaluate it on the SynthScars (Kang et al., 2025) benchmark. To ensure a comprehensive evaluation, we compare our model with advanced closed-sourced and open-sourced image edit methods and artifact correction methods. We assess performance using both subjective and objective evaluations, with results shown in Table 2.

In subjective evaluation, we employ GPT-5.2 (Achiam et al., 2023) as an evaluator, who assesses whether each corrected image contains artifacts related to structure, physics, or distortion. The evaluation scale is binary, with 0 indicating no artifacts and 1 indicating the presence of artifacts. Additionally, we perform human evaluation by randomly selecting 50 images from the test set and obtaining evaluations from 20 volunteers. Each volunteer follows the same criteria to rates the images. The results show that our method outperforms other models across all three artifact categories, achieving lower average artifact scores while maintaining high visual realism. In objective evaluation, we use text-to-image evaluation metrics, with the prompt set uniformly as "A picture taken by a camera." This approach allows us to quantitatively measure the image quality, including perceptual similarity to real-world images, based on various objective scoring functions. Our method achieves the highest scores in HPSv3 (Ma et al., 2025), CLIP-Score (Zhengwentai, 2023), and PickScore (Kirstain et al., 2023), surpassing the second-best results by a significant margin. Fig. 3(a) presents some subjective results of artifact correction. As seen, our method provides detailed and specific anomaly diagnoses and suc-

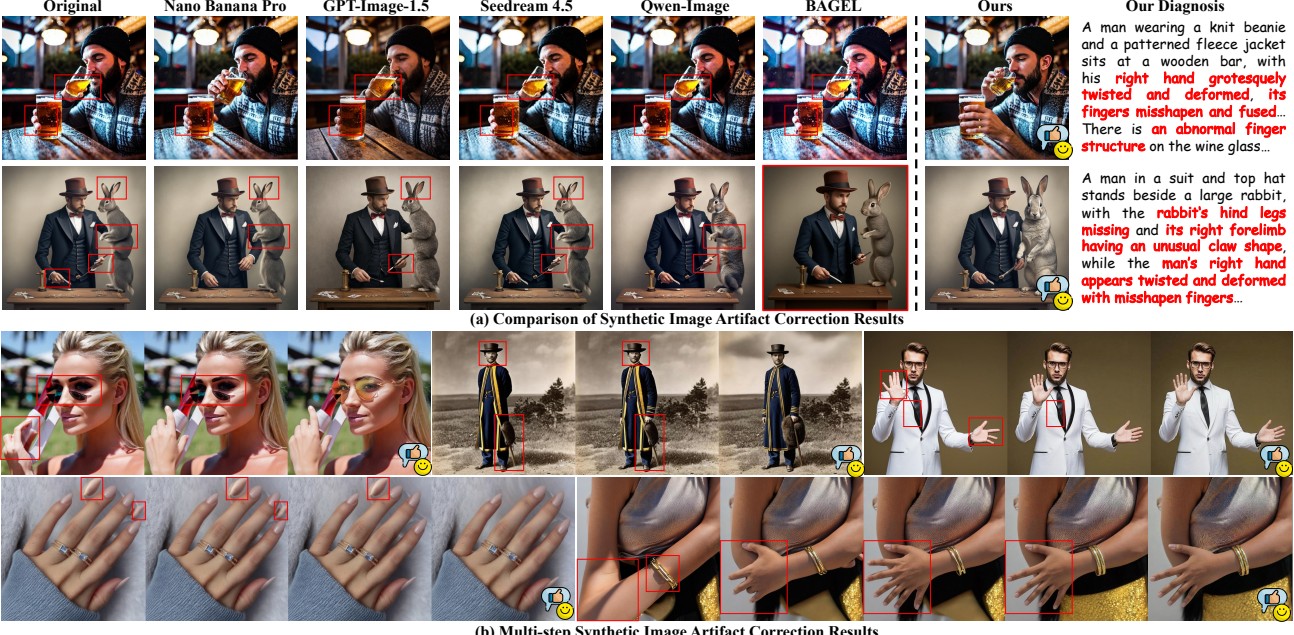

(a) Comparison of Synthetic Image Artifact Correction Results

(b) Multi-step Synthetic Image Artifact Correction Results

(c) Explainable AI-generated Image Detection Result

*Figure 3.* Artifact Correction and Synthetic Detection Results of our GenShield.

cessfully corrects all artifacts. In contrast, even powerful image editing algorithms such as GPT-Image-1.5 are unable to fully correct all anomalies.

Noted that in Table 2, "Ours*" refers to the results obtained by applying a single-step correction, while "Ours" denotes the results from applying multi-step correction. The results demonstrate that multi-step iterative correction significantly enhances both the quality and stability of artifact correction, highlighting the effectiveness of the VCoT strategy in progressively refining images and achieving superior outcomes. Fig. 3(b) shows several examples of multi-step artifact correction. As can be seen, our method progressively eliminates all artifacts through iterative refinement while maintaining the main semantic content of the image.

### 4.4. Ablation Study

To evaluate the impact of different training strategies on our model's performance, we conduct ablation studies with several variants, as shown in Table 3. The first two rows correspond to training with only detection data and only correction data, respectively. The third and fourth rows report the results of directly training with Stage 1 or Stage 2 alone, without the full curriculum progression. The fifth row removes the multi-step VCoT strategy to examine the contribution of iterative self-correction.

Our results demonstrate that optimizing detection or correction tasks independently, or directly training with only Stage 1 or Stage 2, does not achieve the best performance. By combining both detection and correction tasks in Stage 1 and further refining the model in Stage 2, we obtain the strongest overall results. In addition, removing the multi-round VCoT training data leads to a noticeable degradation in correction quality, highlighting the importance of iterative reasoning for artifact removal. This illustrates the effectiveness of our multi-stage joint optimization strategy, which enables better performance across both detection and artifact correction.

### 4.5. Robustness Study of AIGI Detection

Table 4 reports the robustness comparison of AIGI detection methods under these common perturbations. While most baseline detectors suffer noticeable accuracy drops as degradations become stronger, our method remains consistently stable across all settings. In particular, under JPEG compression, we achieve 98.0 and 97.8 accuracy at QF=75 and QF=70, respectively, outperforming the best competing method (AIGI-Holmes, 93.1/92.5) by a large margin. Similar robustness is observed under Gaussian blur, where our accuracy remains 97.5 and 96.9 for $\sigma = 1.0$ and $\sigma = 2.0$, and under resizing ($\times 0.5$) we still achieve 96.1, consistently higher than all baselines. Overall, these

results demonstrate that GenShield provides the most reliable degradation-tolerant detection, which we attribute to the unified understanding–generation training that encourages camera-consistent priors and reduces over-reliance on fragile low-level cues.

## 5. Conclusion

We present GenShield, the first unified autoregressive forensics framework that jointly performs explainable AIGI detection and artifact correction in a curriculum visual CoT mannar, revealing a principled synergy between forensic understanding and generative restoration. To enable rigorous study of this joint task, we construct GenShield-Set, a large-scale, high-quality dataset with precisely aligned artifact–restored image pairs and a unified evaluation pipeline, filling a critical gap left by detection-only benchmarks. Extensive experiments show that GenShield achieves state-of-the-art performance on mainstream AIGI detection benchmarks while outperforming strong open- and closed-source generators in artifact correction quality and generalization to unseen generators.

By unifying diagnosis and repair within a single reasoning-driven framework, this work advances AI image forensics from passive verification toward active authenticity restoration, offering a new paradigm for trustworthy generative systems and a meaningful step toward unified understanding–generation modeling in the AI safety community.

## Impact Statement

This paper presents research aimed at advancing methods for AI-generated image forensics, particularly through unified modeling of detection and artifact correction. While our work may have potential societal implications related to the analysis of synthetic media, we believe these implications are consistent with those commonly encountered in this area of machine learning research and do not warrant separate discussion here.

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

## A. Limitations

Despite its effectiveness, GenShield has several limitations. First, for images with extremely low generation quality—such as those containing large-scale text corruption, severe geometric distortions, or globally incoherent structures—our method may struggle to recover a fully plausible image within a limited number of iterative correction steps. In such cases, the artifact severity exceeds what can be reliably resolved through gradual diagnose–then–correct refinement. Second, the quality and diversity of the correction supervision are inherently bounded by the image editing model used during dataset construction. Since the corrected targets are generated by a strong but imperfect editing model, residual biases or failure patterns in this model may be reflected in the training data and consequently limit the upper bound of correction performance. We expect this limitation to be alleviated as more capable image editing models become available.

## B. More Experimental Results

### B.1. Detection Performance on AIGCDetectBenchmark

We further evaluate our method on the AIGCDetectBenchmark (Zhong et al., 2023), which covers a wide range of image generators, including GAN-based models (e.g., ProGAN (Karras et al., 2017), StyleGAN (Karras et al., 2019), BigGAN (Brock et al., 2018)), diffusion-based models (e.g., Glide (Nichol et al., 2021), Stable Diffusion (Rombach et al., 2022)), as well as recent large-scale generators such as DALL·E 2 (Ramesh et al., 2021) and VQDM (Gu et al., 2022). This benchmark is particularly challenging due to the large diversity of generation mechanisms and the significant domain gap across generators.

As shown in Table 5, our method consistently achieves strong detection performance across almost all generators and attains the best overall mean accuracy of 94.93%, outperforming all competing methods. Notably, our approach maintains near-perfect performance on early GAN-based models such as ProGAN and CycleGAN, while also demonstrating superior robustness on modern diffusion models. For instance, on Glide, our method achieves 98.97% accuracy, significantly surpassing most existing detectors. Compared with strong baselines such as AIDE (Yan et al., 2024), PatchCraft (Zhong et al., 2023), and AIGI-Holmes (Zhou et al., 2025), our method exhibits more balanced and stable performance across generators. While some baselines achieve high accuracy on specific domains (e.g., DIRE-D on ADM-style diffusion models), they often suffer from substantial degradation on others. In contrast, our method avoids such domain-specific bias and delivers consistently high accuracy across both GAN and diffusion based generators, indicating stronger generalization ability.

*Table 5.* **Detection Performance Comparison on the AIGCDetectBenchmark (Zhong et al., 2023) Benchmark.** Accuracy (%) of different detectors (rows) in detecting real and fake images from different generators (columns). The best result and the second-best result are marked in **bold** and underline, respectively.

| Method | ProGAN | StyleGAN | BigGAN | CycleGAN | StarGAN | GauGAN | StyleGAN2 | WFIR | ADM | Glide | Midjourney | SD v1.4 | SD v1.5 | VQDM | Wukong | DALLE2 | Mean |
|---|---|---|---|---|---|---|---|---|---|---|---|---|---|---|---|---|---|
| CNNSpot | **100.00** | 90.17 | 71.17 | 87.62 | 94.60 | 81.42 | 86.91 | 91.65 | 60.39 | 58.07 | 51.39 | 50.57 | 50.53 | 56.46 | 51.03 | 50.45 | 70.78 |
| FreDect | 99.36 | 78.02 | 81.97 | 78.77 | 94.62 | 80.57 | 66.19 | 50.75 | 63.42 | 54.13 | 45.87 | 38.79 | 39.21 | 77.80 | 40.30 | 34.70 | 64.03 |
| Fusing | **100.00** | 85.20 | 77.40 | 87.00 | 97.00 | 77.00 | 83.30 | 66.80 | 49.00 | 57.20 | 52.20 | 51.00 | 51.40 | 55.10 | 51.70 | 52.80 | 68.38 |
| LNP | 99.67 | 91.75 | 77.75 | 84.10 | 99.92 | 75.39 | 94.64 | 70.85 | 84.73 | 80.52 | 65.55 | 85.55 | 85.67 | 74.46 | 82.06 | 88.75 | 83.84 |
| LGrad | 99.83 | 91.08 | 85.62 | 86.94 | 99.27 | 78.46 | 85.32 | 55.70 | 67.15 | 66.11 | 65.35 | 63.02 | 63.67 | 72.99 | 59.55 | 75.34 | 75.34 |
| UnivFD | 99.81 | 84.93 | 95.08 | 98.33 | 95.75 | **99.47** | 74.96 | 86.90 | 66.87 | 62.46 | 56.13 | 63.66 | 63.49 | 85.31 | 70.93 | 50.75 | 78.43 |
| DIRE-G | 95.19 | 83.03 | 70.12 | 74.19 | 95.47 | 67.79 | 75.31 | 58.05 | 75.78 | 71.75 | 58.01 | 49.74 | 49.83 | 53.68 | 54.46 | 66.48 | 68.68 |
| DIRE-D | 52.75 | 51.31 | 49.70 | 49.58 | 46.72 | 51.23 | 51.72 | 53.30 | **98.25** | 92.42 | 89.45 | 91.24 | 91.63 | 91.90 | 90.90 | 92.45 | 71.53 |
| PatchCraft | **100.00** | 92.77 | 95.80 | 70.17 | **99.97** | 71.58 | 89.55 | 85.80 | 82.17 | 83.79 | 90.12 | **95.38** | **95.30** | 88.91 | 91.07 | **96.60** | 89.31 |
| NPR | 99.79 | 97.70 | 84.35 | 96.10 | 99.35 | 82.50 | 98.38 | 65.80 | 69.69 | 78.36 | 77.85 | 78.63 | 78.89 | 78.13 | 76.11 | 64.90 | 82.91 |
| AIDE | 99.99 | **99.64** | 83.95 | **98.48** | 99.91 | 73.25 | 98.00 | 93.43 | 94.20 | 95.09 | 77.20 | 93.00 | 92.85 | **95.16** | **93.55** | 96.60 | 92.77 |
| AIGI-Holmes | **100.00** | 98.35 | 94.51 | 97.03 | **100.00** | 95.19 | **98.88** | 95.71 | 88.43 | 91.53 | 81.56 | 91.28 | 91.38 | 90.94 | 89.46 | 85.32 | 93.16 |
| **Ours** | **100.00** | 97.39 | **96.37** | 97.60 | 99.10 | 98.60 | 97.73 | 94.11 | 94.89 | **98.97** | **91.15** | 92.03 | 89.95 | 91.30 | 92.63 | 87.09 | **94.93** |

### B.2. More Subjective Results

**VCoT Correction Results.** To better illustrate the effectiveness of our VCoT-based artifact correction, we present representative qualitative examples in Fig. 6, 7. As shown, the model performs multi-round, alternating diagnosis and

correction: it first describes the suspicious regions in a structured manner and then applies targeted edits to progressively remove artifacts while preserving the main semantics. Importantly, the process is adaptive—when the image is sufficiently corrected, the model outputs a termination-style diagnosis indicating that no obvious artifacts remain, and stops further correction automatically.

**AIGI Detection Results.** To better demonstrate the effectiveness of our explainable AIGI detection, we present representative qualitative examples in Fig. 8, 9. As shown, our model outputs a structured prediction together with detailed rationales, analyzing the image from multiple forensic perspectives such as geometry and perspective consistency, lighting and shadow coherence, texture and edge artifacts, and local distortions. These explanations further highlight visually suspicious regions and provide evidence-based cues that support the final real/fake decision.

### B.3. Distribution Analysis of Correction Steps

Fig. 4 shows the distribution of the actual number of correction iterations required in the test set. We observe that the majority of samples can be successfully repaired within only a few editing steps: about 79.8% of images terminate after two rounds, while 11.5% require three rounds and 6.6% require four rounds. Only a very small fraction of difficult cases need more iterations, with 1.8% finishing in five steps and merely 0.3% reaching six steps. This distribution demonstrates that our iterative VCoT correction process is both efficient and adaptive, converging quickly for most inputs while still being capable of handling more challenging artifacts through additional refinement.

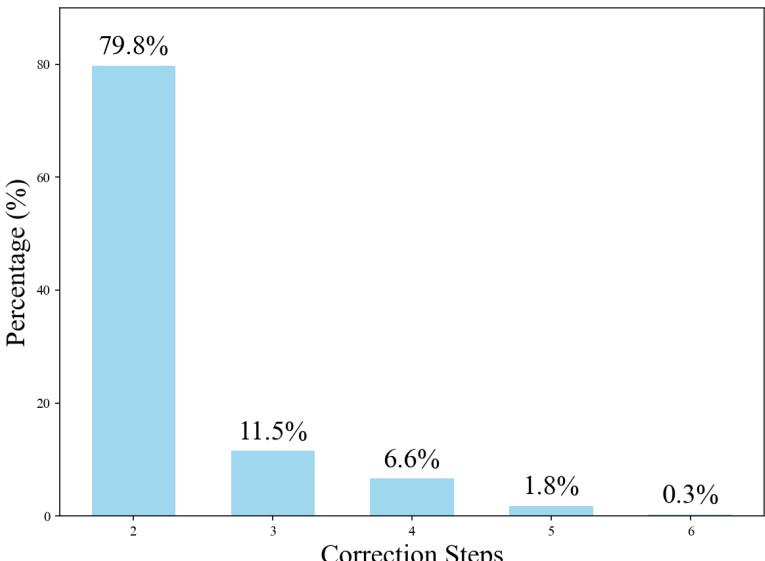

*Figure 4.* **Distribution of correction steps on the test set.** Most images are repaired within 2–4 iterative editing rounds, while only a few require more steps.

### B.4. Reproduced Detection Performance of Non-LLM Baselines

To further improve the transparency and reproducibility of the comparison in Table 1, we additionally reproduce the non-LLM-based detector baselines under the same Holmes-Set evaluation protocol used in the main paper. Specifically, we follow the official implementations and training/evaluation settings adopted by AIGI-Holmes, and retrain the non-LLM detectors on Holmes-Set for AI-generated image detection. The reproduced results are reported in Table 6. Although small numerical variations can occur due to training stochasticity, implementation details, and runtime environments, the reproduced results are overall consistent with the performance level reported in the main comparison. More importantly, these reproduced results do not change the main conclusion of Table 1: GenShield consistently achieves stronger detection performance than existing non-LLM detector baselines across diverse image generators. We include these results to remove any ambiguity about the experimental protocol and to facilitate future reproducibility.

*Table 6.* Reproduced results of non-LLM-based detector baselines on Holmes-Set (Zhou et al., 2025) under the same evaluation protocol as Table 1. All results are reported in accuracy (%). For reference, we also include the **Ours** results from Table 1 in the last row.

| Method | Janus | Janus-Pro-1B | Janus-Pro-7B | Show-o | LlamaGen | Infinity | VAR | PixArt-XL | SD3.5-Large | FLUX | Mean |
|---|---|---|---|---|---|---|---|---|---|---|---|
| CNNSpot | 77.4 | 67.7 | 87.2 | 80.7 | 65.7 | 91.0 | 61.5 | 78.3 | 61.0 | 81.9 | 75.2 |
| AntifakePrompt | 74.1 | 89.9 | 91.9 | 84.1 | 94.1 | 90.8 | 89.8 | 84.4 | 91.3 | 61.8 | 85.2 |
| UnivFD | 85.8 | 94.0 | 92.5 | 82.4 | 86.3 | 83.3 | 62.5 | 78.4 | 92.0 | 73.4 | 83.1 |
| NPR | 64.6 | 66.8 | 81.3 | 93.9 | 94.2 | 97.2 | 88.8 | 98.0 | 94.5 | 93.2 | 87.2 |
| LaRE | 65.9 | 73.6 | 97.7 | 76.8 | 92.3 | 71.2 | 94.5 | 79.3 | 90.1 | 79.1 | 82.1 |
| RINE | 91.9 | 99.0 | 99.1 | 97.7 | **99.0** | **99.8** | 87.5 | 98.9 | 97.5 | 96.9 | 96.7 |
| AIDE | 93.0 | 92.5 | 96.0 | 95.3 | 98.8 | 98.6 | 94.7 | 98.8 | **98.9** | 95.2 | 96.2 |
| **Ours** | **98.8** | **99.5** | **99.4** | **99.2** | 98.8 | 98.9 | **99.4** | **99.3** | 96.7 | **97.9** | **98.8** |

## B.5. Additional Ablation Studies on Detection-Correction Synergy

To further examine the synergy between AIGI detection and artifact correction, we provide additional ablation studies from both the detection and correction perspectives. For detection, we evaluate the model on unseen advanced closed-source generators from the AIGI-Now dataset (Chen et al., 2025b), including GPT-4o, JiMeng, Keling, and MiniMax. For correction, we adopt broader evaluation metrics, including image quality, aesthetics, and human preference. As shown in Table 7, the full GenShield model consistently outperforms the single-task variants. Compared with the detection-only model, joint training with correction improves the average detection accuracy from 87.8% to 93.4%. Compared with the correction-only model, joint training with detection also improves all correction-related metrics. These results provide additional empirical evidence that detection and correction benefit from being jointly modeled in our unified framework.

*Table 7.* Additional ablation studies on the synergy between AIGI detection and artifact correction. For detection, we report accuracy (%) on unseen closed-source generators from AIGI-Now. For correction, we report image quality, aesthetics, and human preference metrics.

| Method | AIGI Detection | | | | | Artifact Correction | | | |
|---|---|---|---|---|---|---|---|---|---|
| | GPT-4o | JiMeng | Keling | MiniMax | Mean | Q-Insight ↑ | Aesthetic ↑ | ImageReward ↑ | HPSv2.1 ↑ |
| Only Det. | 85.4 | 82.9 | 88.1 | 94.6 | 87.8 | – | – | – | – |
| Only Corr. | – | – | – | – | – | 3.12 | 5.17 | -0.37 | 19.61 |
| **Ours** | **92.2** | **88.3** | **94.3** | **98.6** | **93.4** | **3.89** | **6.34** | **0.23** | **22.15** |

# C. Details of Artifact Correction Evaluation Pipeline

Since there is no widely adopted and comprehensive benchmark protocol for evaluating AI image artifact correction, we build a unified evaluation pipeline that measures both artifact removal quality and realistic realism of corrected images. Following the setting in Sec. 4.3, our pipeline consists of two complementary components: subjective evaluation (artifact-oriented judgments) and objective evaluation (realistic preference and text–image alignment).

## C.1. Subjective Evaluation

We design a structured, reproducible subjective evaluation protocol to assess whether corrected images still contain common synthetic artifacts. Based on the artifact taxonomy (Kang et al., 2025) used in our refinement dataset, we categorize artifacts into three groups: Structure (anatomy and geometry), Physics (lighting and interaction), and Distortion (texture and signal). To reduce scoring ambiguity and mitigate evaluator bias introduced by fine-grained rating scales, we adopt a binary scoring rule for each category: 0 indicates no obvious artifacts and 1 indicates artifacts present.

**GPT-assisted evaluation.** For scalability and reproducibility, we employ GPT-5.2 (OpenAI, 2025) as an automatic evaluator. We prompt the evaluator to judge the authenticity of each corrected image by checking artifacts along the three categories above, and output a structured result with three binary scores. The full prompt template is shown in Fig. 5. It provides domain knowledge and explicit criteria for each category (e.g., geometric consistency and text legibility for Structure; shadow/reflection consistency and gravity/contact for Physics; over-smoothed textures, repetitive patterns, and unnatural noise for Distortion), and enforces a strict output format to avoid free-form explanations.

**Human evaluation.** To further validate the reliability of GPT-based judgments, we conduct a human study under the same

# Task
Evaluate the input image to determine its visual authenticity (Real vs. AI-generated). You must analyze the image based on 3 specific dimensions using a strict 3-point scale.

# Domain Knowledge & Evaluation Criteria

## 1. Structure (Anatomy & Geometry)
Knowledge: Real objects follow strict biological and geometric laws. AI models often struggle with complex coherency.
What to check:  Anatomy: Hand/finger count, joint articulation, eye symmetry (pupil shape/iris), limb connections.
Geometry: Straight lines on buildings, perspective vanishing points, object symmetry, and text legibility.
Scoring: 0.0: has no artifacts, 1.0: has artifacts

## 2. Physics (Lighting & Interaction)
Knowledge: Light travels in straight lines. Shadows and reflections must mathematically map to the environment and light sources.
What to check:  Lighting: Consistency of light direction across all objects.
Shadows: Presence, direction, and hardness/softness matching the light source.
Reflections: Accuracy in mirrors, water, or eyes (must match the scene).
Gravity: Objects should have weight and contact with the ground.
Scoring: 0.0: has no artifacts, 1.0: has artifacts

## 3. Distortion (Texture & Signal)
Knowledge: Real photographs contain optical imperfections (lens blur, ISO noise) and organic irregularities. AI often generates "too clean" or "smeared" textures.
What to check:  Texture: Skin pores vs. "plastic/waxy" smoothing, hair separation, fabric weave.
Artifacts: Digital smearing, strange color bleeding, repetitive patterns.
Noise: Natural film grain/sensor noise vs. clean digital silence.
Scoring: 0.0: has no artifacts, 1.0: has artifacts

# Output Rules
Return ONLY the result string in the specific format below.
NO explanation, NO analysis text, NO markdown formatting (like ``` code blocks).
Just the raw string.

# Output Format
<Structure>[Score]</Structure><Physics>[Score]</Physics><Distortion>[Score]</Distortion>

*Figure 5.* **GPT-assistant Evaluation Prompt.**

rules. We randomly sample 50 images from the test set and collect ratings for all methods on these samples. We invite 20 volunteers to independently score each corrected result using the same binary criteria for Structure/Physics/Distortion. We report average human scores in Tab. 2, and observe consistent trends with the GPT-assisted evaluation, supporting the robustness of our subjective pipeline.

## C.2. Objective Evaluation

In addition to subjective judgments, we evaluate corrected images with objective metrics to quantify realistic realism and preference alignment. We follow recent image generation evaluation practice and report HPSv3 (Ma et al., 2025) and PickScore (Kirstain et al., 2023), which are trained on large-scale human preference data, as well as CLIPScore (Zhengwentai, 2023), which measures text-image alignment. All three metrics require a text prompt; to avoid method-specific prompt engineering and to ensure a uniform target across all corrected outputs, we fix the prompt to: "A picture taken by a camera." This prompt explicitly encodes the desired realistic distribution and provides a consistent reference for comparing different correction methods. Together with the subjective artifact judgments, our pipeline enables a comprehensive and reproducible evaluation of artifact correction quality.

# D. Details of GenShield-Set

As illustrated in Fig. 1, we build GenShield-Set to support the joint training of explainable detection and synthetic artifact correction. It contains two complementary subsets: **GenShield-Set-Detect** for detection and rationale generation, and **GenShield-Set-Correct** for instruction-guided artifact correction and iterative VCoT refinement.

## D.1. GenShield-Set-Correct

**Prompt-enhanced correction target generation.** We construct GenShield-Set-Correct using anomalous images and their anomaly-region textual annotations from SynthScars (Kang et al., 2025) (Fig. 1(a)). For each anomalous image, we perform repair prompt enhancement by rewriting the raw region description into a standardized instruction that (i) explicitly specifies the abnormal area and abnormal pattern, and (ii) enforces a realistic, anomaly-free target while preserving identity and global content. We employ an advanced proprietary image editing model to generate corrected image candidates.

**Expert filter for high-quality restorations.** Automatic correction may produce noisy outputs and failure cases that can bias training. To ensure data quality, we apply an expert-based repaired image filter (Fig. 1(a)), where 20 trained annotators are instructed to remove candidates with common failure modes, including: (i) incomplete/failed restoration (artifacts remain visible or correction is insufficient), (ii) artifact introduction (new unnatural patterns are created), (iii) content drift (identity/object attributes are changed), (iv) excessive detail loss (over-smoothing or texture collapse), (v) structural breakage (geometry/anatomy becomes inconsistent) and (vi) unnatural seams (local edits mismatch surrounding context). Each candidate is reviewed by three experts, and we keep it only if at least two experts agree that it is artifact-reduced and semantically consistent with the original. After filtering, we obtain over 10K high-quality corrected images, forming paired supervision tuples for training.

**Varying repair completeness for multi-step refinement.** To support multi-step VCoT correction and increase training diversity, we additionally construct varying repair completeness images (Fig. 1(a)). Specifically, we use our Stage 1 checkpoint to perform instruction-guided correction and retain outputs. These partially corrected images are then treated as the starting point of restoration in Stage 2, enabling the model to learn iterative diagnose–correct behaviors from different correction states rather than only from the original anomalous inputs.

**Termination-style supervision.** As shown in Fig. 1(a), we also create termination-style correction answers for clean or sufficiently corrected images. The model is supervised to explicitly output that no artifacts are observed and the image already appears normal. This supervision is later used to train an explicit stopping behavior in iterative VCoT correction.

## D.2. Correction Data Examples

To illustrate the structure of our correction data, we randomly sample several examples from GenShield-Set-Correct, as shown in Fig. 10. Each example contains (i) an anomalous input image ($\mathbf{I}_{\text{artifact}}$ with visible artifacts, (ii) an initial diagnostic description $\mathbf{T}_{\text{diag}}$ that summarizes the artifact type and indicates the suspicious region, (iii) the corresponding corrected image $\mathbf{I}_{\text{correct}}$ after artifact removal, and (iv) a termination diagnosis $\mathbf{T}_{\text{stop}}$ generated on the corrected image, where the model explicitly states that no artifacts are observed and the image appears normal.

# E. More Implementation Details

Table 8 summarizes the full training recipe of GenShield, including the two-stage curriculum and the ablation settings (Only Det. and Only Corr.). Across all settings, we adopt a constant learning rate of $2 \times 10^{-5}$ with zero weight decay, gradient-norm clipping of 1.0, and AdamW optimizer ($\beta_1 = 0.9$, $\beta_2 = 0.95$, $\epsilon = 1.0 \times 10^{-15}$). We use 500 warm-up steps for all runs. Stage 1 and Stage 2 are each trained for 5K steps, while the Only Det. and Only Corr. ablations are trained for 10K steps to match the overall update budget. We maintain the same loss weighting between the text cross-entropy loss and the image reconstruction loss with a ratio of 0.25:1 for both Stage 1 and Stage 2. Exponential moving average is applied with EMA ratios of 0.999 for Stage 1 and 0.990 for Stage 2, and 0.990 for both Only Det. and Only Corr. For data preprocessing, correction inputs are randomly resized with the minimum short side and maximum long side in $(512, 1024)$, while detection inputs are resized to $(378, 980)$. We also apply a diffusion timestep shift of 4.0 for the correction branch. The bottom part of Table 8 lists the data sampling ratios used in each setting: Stage 1 jointly samples AIGI detection and instruction-guided artifact correction with a 1.0:5.0 ratio; Stage 2 increases the detection sampling to 2.0 and mixes multiple VCoT correction trajectories with ratios 1.0 (Initial state), 1.0 (Intermediate state), and 0.1 (Terminate state); Only Det.

trains solely on detection data; and Only Corr. trains solely on VCoT correction data (Initial state).

*Table 8.* **Training recipe of GenShield.**

| | Stage 1 | Stage 2 | Only Det. | Only Corr. |
|---|---|---|---|---|
| **Hyperparameters** | | | | |
| Learning rate | $2 \times 10^{-5}$ | $2 \times 10^{-5}$ | $2 \times 10^{-5}$ | $2 \times 10^{-5}$ |
| LR scheduler | Constant | Constant | Constant | Constant |
| Weight decay | 0.0 | 0.0 | 0.0 | 0.0 |
| Gradient norm clip | 1.0 | 1.0 | 1.0 | 1.0 |
| Optimizer | \multicolumn{4}{c}{AdamW ($\beta_1 = 0.9$, $\beta_2 = 0.95$, $\epsilon = 1.0 \times 10^{-15}$)} | | | |
| Loss weight (CE : MSE) | 0.25 : 1 | 0.25 : 1 | - | - |
| Warm-up steps | 500 | 500 | 500 | 500 |
| Training steps | 5K | 5K | 10k | 10K |
| EMA ratio | 0.999 | 0.990 | 0.990 | 0.990 |
| Corr. resolution (min short side, max long side) | (512, 1024) | | | |
| Det. resolution (min short side, max long side) | (378, 980) | | | |
| Diffusion timestep shift | 4.0 | | | |
| **Data sampling ratio** | | | | |
| AIGI Detection | 1.0 | 2.0 | 1.0 | 0.0 |
| Instruction Guided Artifact Correction | 5.0 | 0.0 | 0.0 | 0.0 |
| Artifact Correction with VCoT (Initial state) | 0.0 | 1.0 | 0.0 | 1.0 |
| Artifact Correction with VCoT (Intermediate state) | 0.0 | 1.0 | 0.0 | 0.0 |
| Artifact Correction with VCoT (Terminate state) | 0.0 | 0.1 | 0.0 | 0.0 |

# F. Comparison Methods

With the continuous development of deep learning (Zhou et al., 2026; Li et al., 2025b; Zhang et al., 2026b; Wu et al., 2024; Li et al., 2025c), especially large language models (Chen et al., 2024; Hu et al., 2026; Zhou et al.; Liu et al., 2026; Wu et al., 2026) and generative models (Li et al., 2026b; Wu et al., 2025b; Jiang et al., 2026), existing methods have explored diverse technical routes for both synthetic image detection and artifact correction. For AI synthetic image detection, prior works range from low-level artifact modeling, frequency-domain analysis, gradient-based representations, reconstruction errors, and texture cues to semantic reasoning and multimodal/LLM-based explainable detection. For AI synthetic image artifact correction, recent approaches mainly rely on powerful image generation or editing models, including closed-source commercial systems, open-source editing models, and unified multimodal generation frameworks. To provide a comprehensive comparison, we include representative baselines from both directions and briefly introduce them below.

## F.1. AI Synthetic Image Detection Method

**CNNSpot.** (Wang et al., 2020) CNNSpot builds a forgery detector based on ResNet and finds that data augmentations such as JPEG compression and Gaussian blur can effectively improve the model's generalization to unseen generative architectures and data sources.

**AntifakePrompt.** (Chang et al., 2023) AntifakePrompt formulates fake image detection as a visual question answering task, leveraging prompt-tuned vision-language models to distinguish real from generated images with zero-shot capability and minimal additional parameters.

**UniFD.** (Ojha et al., 2023) UniFD trains a detector using the feature space extracted by a large pre-trained vision-language model (CLIP: ViT-L/14). The use of a large pre-trained model results in smoother decision boundaries, which enhances the generalization ability of the detector.

**NPR.** (Tan et al., 2024) NPR captures and models local pixel dependencies introduced by upsampling operations in CNN-based generative networks to identify universal forgery traces from various generative models, thereby improving the generalization performance of deepfake detection.

**LaRE.** (Luo et al., 2024) LaRE improves diffusion image detection (e.g., DIRE (Wang et al., 2023)) by using latent reconstruction error as the core feature and introducing an error-guided refinement module (EGRE) to optimize features across spatial and channel dimensions, boosting performance and achieving 8× speedup.

**RINE.** (Koutlis & Papadopoulos, 2024) RINE leverages fine-grained features from intermediate CLIP Transformer blocks, maps them into a forgery-aware space via a lightweight network, and uses a learnable module to weight block importance. It is optimized with cross-entropy and contrastive losses, significantly enhancing synthetic image detection and generalization.

**AIDE.** (Yan et al., 2024) AIDE concatenates DCT-based local frequency features with CLIP global semantics, leveraging multi-expert extraction to capture both low-level artifacts and high-level cues, achieving effective AI-generated image detection and outperforming existing methods on benchmarks.

**FreDect.** (Frank et al., 2020) FreDect identifies deepfake images by detecting distinct frequency-domain artifacts caused by GAN-generated images. This method leverages abnormal frequency patterns to effectively recognize fake images.

**Fusing.** (Ju et al., 2022) Fusing presents a two-branch model that combines global and local features to enhance the generalization of AI-synthesized image detection, using multi-head attention to improve the accuracy across various models and image resolutions.

**LNP.** (Liu et al., 2022) LNP introduces a method for detecting generated images by focusing solely on real images, overcoming common issues in efficiency and generalization, and showing robustness against post-processing with 99.9% less training data.

**LGrad.** (Tan et al., 2023) The paper introduces a new detection framework called LGrad, which uses gradient learning to detect GAN-generated images by training a transformation model to convert images into a specific gradient space, effectively identifying fake images with reduced training data.

**DIRE.** (Wang et al., 2023) DIRE distinguishes real images from diffusion-generated ones by measuring the error between the input image and its reconstruction from a pre-trained diffusion model, based on the observation that diffusion-generated images can be approximately reconstructed by the diffusion model, whereas real images cannot.

**PatchCraft** (Zhong et al., 2023) PatchCraft introduces an AI-generated image detection method that focuses on texture patches, enhancing the detection of fake images by leveraging the contrast between rich and poor texture regions within an image.

**AIGI-Holmes.** (Zhou et al., 2025) AIGI-Holmes builds on the Holmes-Set with instruction tuning and human preferences, using a three-stage pipeline that feeds CLIP and NPR features into LLaVa with LoRA fine-tuning. A collaborative decoding strategy fuses vision and semantics, enabling interpretable and generalizable AI-generated image detection.

**FakeVLM.** (Wen et al., 2025) FakeVLM is a multimodal model for synthetic image detection, trained on the 100K-image FakeClue dataset with fine-grained textual annotations. It detects forgeries and generates natural language explanations, achieving high performance and interpretability without extra classifiers.

### F.2. AI Synthetic Image Artifact Correction Method

**GPT-Image-1.5.** (OpenAI, 2025) GPT-Image-1.5 is a flagship image generation model that improves on its predecessor with better prompt understanding, more precise edits, enhanced text rendering, and significantly faster image generation—delivering high-quality, photorealistic outputs in up to 4× less time.

**Nano-Banana.** (Comanici et al., 2025) Nano-Banana is a lightweight AI image generation and editing model designed for fast, efficient creative prototyping, supporting multi-round editing consistency and multi-image fusion. With low latency and cost, it is ideal for drafts, social media content, and rapid ideation.

**Nano-Banana-Pro.** (Comanici et al., 2025) Nano-Banana-Pro is the advanced version built on the Gemini 3 Pro architecture, supporting 4K resolution, accurate multilingual text rendering, and fusion of up to 14 reference images with consistent portrayal of up to 5 individuals. It offers professional-grade controls (e.g., lighting, camera angles), making it suitable for commercial content production and branded assets.

**Seedream 4.5.** (Seedream et al., 2025) Seedream is an efficient next-generation multimodal image generation system that unifies text-to-image generation, image editing, and multi-image composition through a high-performance Diffusion Transformer + VAE architecture. It can rapidly produce 1K–4K high-resolution images (with 2K image inference in just

1.8 seconds), offering strong generalization and precise multimodal reasoning for complex tasks—making it a powerful interactive tool for both creative and professional applications.

**FLUX-Pro.** (Labs, 2024) FLUX-Pro is a flagship text-to-image generation model, offering state-of-the-art inference speed, strong prompt adherence, and high visual quality, making it ideal for efficient, scalable AI-based image creation via API.

**BAGEL.** (Deng et al., 2025) BAGEL is an open-source unified multimodal foundation model that adopts a decoder-only architecture. It is pre-trained on trillions of tokens of interleaved multimodal data, including text, images, and videos, and significantly outperforms other open-source models of the same kind in multimodal understanding/generation and complex reasoning tasks (such as free-form image manipulation, 3D manipulation, etc.).

**Qwen-Image-Edit-2511.** (Wu et al., 2025b) Qwen-Image-Edit-2511 is an enhanced AI image editing model that greatly improves character and multi-subject consistency over its predecessor, enabling stable natural-language-driven edits, identity preservation, and multi-image fusion. It also integrates built-in community LoRA capabilities and strengthened geometric reasoning for high-fidelity, production-ready editing.

**Step1X-Edit-v1p2.** (Liu et al., 2025) Step1X-Edit-v1p2 is an advanced open-source model for general image editing, combining a multimodal LLM with a diffusion decoder to generate target images based on reference images and natural language instructions. It achieves state-of-the-art performance on the newly introduced GEdit-Bench, significantly surpassing existing open-source methods and approaching the performance of closed models like GPT-4o and Gemini Flash.

You are an image anomaly repair assistant. This is an AI-generated image, but it has some defects and anomalies. Please fix these abnormal areas so that it looks as realistic and natural as possible.

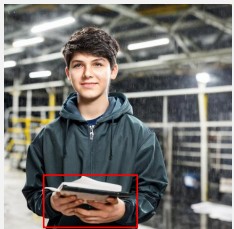

[GT: The boy had an abnormal hand structure and the wrong number of fingers.]

**One-turn**

<diagnosis> Upon examining the image. I have found: A young man in a warehouse setting holds a book, with his left hand showing a deformed thumb and a missing index finger, while his right hand has a deformed thumb and a missing index finger.\nTo elaborate, I have found the following artifacts. Boys left hand:The boy's left thumb is deformed and bent, and his index finger is missing.Boys right hand:The boy's right thumb is deformed and bent, and his index finger is missing. </diagnosis> <repair>

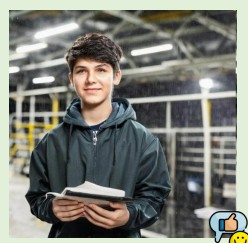

**Two-turn (Stop)**

<diagnosis> Upon examining the image, I have found: a young man with short dark hair and a slight smile, wearing a dark green hooded jacket. He is holding a notebook and pen in his hands and standing in what appears to be an industrial or warehouse setting with visible shelving and overhead lights. The environment suggests a rainy day, as there are visible raindrops in the air.\nTo elaborate, I don't notice any artifacts, this image is already a normal image. </diagnosis> <repair>

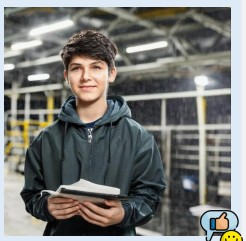

**One-turn**

<diagnosis> Upon examining the image. I have found: A cute kitten is seated on a green surface and gazes directly forward with slightly unusual-looking eyes. Its right front paw appears deformed and displays an extra toe.\nTo elaborate, I have found the following artifacts. Kitten's unusual-looking eyes and Kitten's right front paw:The kitten's right front paw is deformed and has an extra toe. </diagnosis> <repair>

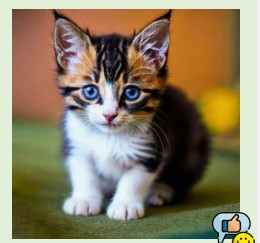

**Two-turn (Stop)**

<diagnosis> Upon examining the image, I have found: a close-up portrait of a calico kitten with striking blue eyes and a mix of white, black, and orange fur. The kitten is sitting on a green surface, looking directly at the camera with a curious expression. The background is softly blurred, featuring warm orange and yellow tones.\nTo elaborate, I don't notice any artifacts, this image is already a normal image. </diagnosis> <repair>

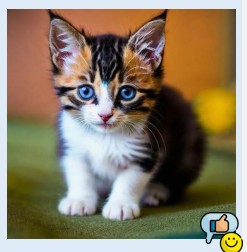

[GT: The cat's eyes and paws are abnormal.]

[GT: The pickup truck had abnormal wheels, bumpers, mirrors, and windshields.]

**One-turn**

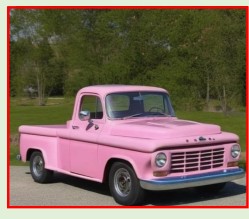

<diagnosis> Upon examining the image. I have found: A vintage pink pickup truck parked on a road. The right rearview mirror is missing, the wheels appear deformed and twisted, the glass material looks abnormal, and the bumper structure is irregular.\nTo elaborate, I have found the following artifacts. The truck is missing its right rearview mirror. The wheels are visibly deformed and twisted. The glass material appears unrealistic, and the bumper structure is abnormal, all of which are inconsistent with the structure of a real vehicle.</diagnosis> <repair>

**Two-turn**

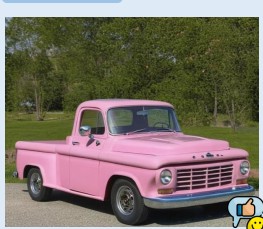

<diagnosis> Upon examining the image, I have found: a classic pink pickup truck with a retro design, featuring smooth, rounded body lines, a large front grille, and chrome detailing. It is positioned on a paved road, with a backdrop of green grass and leafy trees under a clear sky.\nTo elaborate, the image appears blurry and heavily compressed. </diagnosis> <repair>

**Three-turn (Stop)**

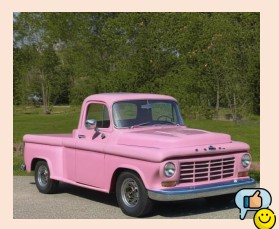

<diagnosis> Upon examining the image, I have found: a vintage pink pickup truck parked on a paved surface. The vehicle features a classic design with rounded edges, a prominent front grille, and chrome accents. The background consists of a grassy field and a line of trees under a clear sky.\nTo elaborate, I don't notice any artifacts, this image is already a normal image.</diagnosis> <repair>

[GT: The woman's hand structure, sunglasses lenses, and eye parts are abnormal.]

**One-turn**

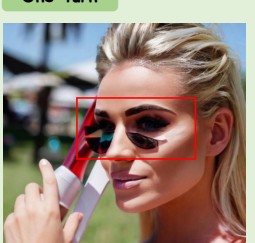

<diagnosis> Upon examining the image. I have found: A woman with blonde hair and sunglasses holds a red and white object, with her left hand showing a deformed thumb and a missing finger.\nTo elaborate, I have found the following artifacts. The character's left hand:The character's left thumb is deformed and bent, and a finger is missing, which is significantly different from the shape of a normal finger. </diagnosis> <repair>

**Two-turn**

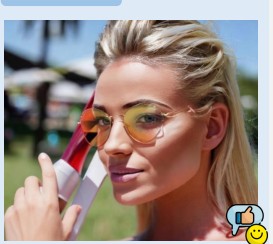

<diagnosis> Upon examining the image. I have found: Sunglasses on a woman's face are unusual, the left lens is missing. Additionally, the area around the woman's eyes appears slightly distorted.\nTo elaborate, I have found the following artifacts. the left lens of the sunglasses is absent and not connected to the right lens, breaking the structural integrity of the eyewear. The woman's eye contours are also unclear, contributing to an overall unrealistic appearance in that region. </diagnosis> <repair>

**Three-turn (Stop)**

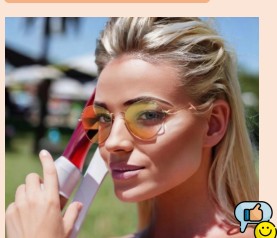

<diagnosis> Upon examining the image, I have found: a high-quality portrait of a young woman with blonde hair and striking makeup. She is wearing reflective sunglasses and holding a red and white object close to her face. The background is a blurred outdoor scene with greenery and structures, suggesting a sunny day.\nTo elaborate, I don't notice any artifacts, this image is already a normal image. </diagnosis> <repair>

*Figure 6.* **Artifact Correction Results of our GenShield.**

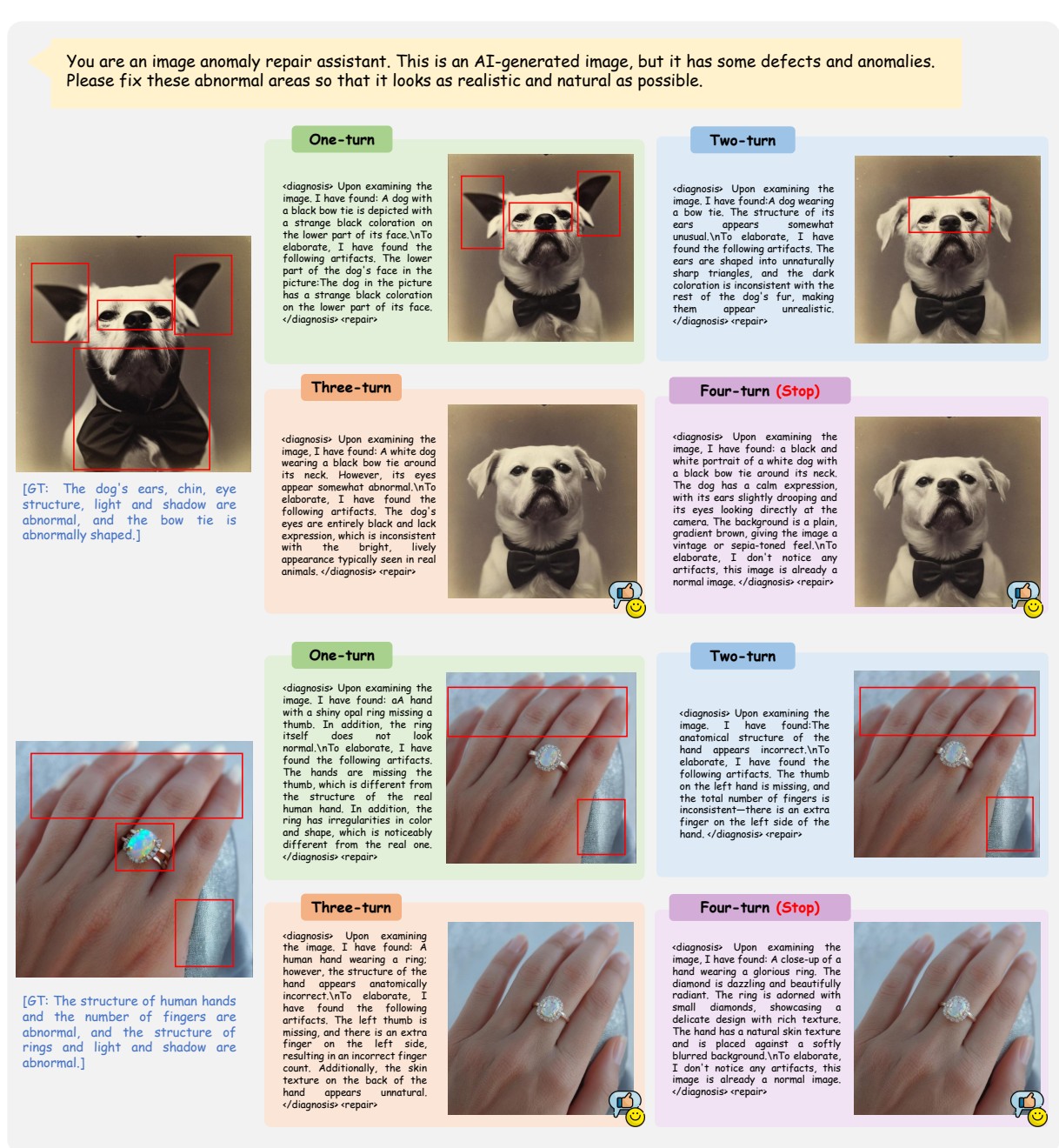

*Figure 7.* **More Artifact Correction Results of our GenShield.**

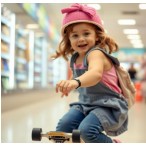

Please evaluate whether this image is an AI creation or something real, and provide an explanation.

[GT: Fake]

**<detect>**This is a fake image. **</detect>**
**<caption>**Image Description: The image depicts a young girl riding a skateboard in what appears to be a supermarket aisle. She is wearing a pink hat with a bow, a pink tank top, denim overalls, and a backpack. The girl is smiling and looking directly at the camera. The background shows shelves stocked with various products, and the lighting suggests an indoor setting. **</caption>**
**<reason>**Based on the provided fake image, here are the reasons why this image is fake:
1. Line segments: The shadows in the image do not align with the light source. The girl's shadow on the skateboard and the floor is inconsistent with the direction of the light, suggesting that the image may have been manipulated. Additionally, the parallel lines of the shelves in the background do not converge at a single vanishing point, which is unnatural for a real photograph.
2. Edges: There are noticeable discontinuities and inconsistencies in the edges, particularly around the girl's arms and the skateboard. The edges appear overly sharp in some areas and blurred in others, which is a common artifact in AI-generated images.
3. Texture: The texture of the girl's clothing, especially the denim overalls, appears unnaturally smooth and lacks the detailed texture of real denim. Additionally, the background products on the shelves have a repetitive and blurry texture pattern, which is not typical in real photographs.
4. Distortion: The girl's hands and the skateboard exhibit slight distortions that do not conform to the natural shapes of real objects. The skateboard, in particular, appears to be slightly warped, which is not typical for a real skateboard.
5. Overall Hue: The overall tone of the image is overly vibrant, especially in the colors of the girl's clothing and the background. This hyper-realistic coloring is a common characteristic of AI-generated images.
6. Clarity: The image shows signs of unnatural pixel blurriness, particularly in the background and around the edges of the girl's clothing. This suggests that the image may have been generated or manipulated by AI.
7. Perspective: The perspective of the image is inconsistent. The girl appears to be riding the skateboard at an unnatural angle, and the depth of field does not change reasonably between the foreground and background.
8. Shadows: The shadows in the image are not consistent with the light source. The girl's shadow on the skateboard and the floor is not aligned with the direction of the light, which is a common flaw in AI-generated images.
9. Faces: The girl's facial features appear overly perfect and lack the natural imperfections of a real face. The edges of her face and facial features are slightly blurred and distorted, which is a common artifact in AI-generated images.
10. Body Structure: The girl's body structure exhibits some anatomical errors. Her arms and hands appear disproportionately large compared to her body, and her posture while riding the skateboard is unnatural.
11. Common Sense: It is highly unusual for a young girl to be riding a skateboard in a supermarket aisle, which violates common sense and typical behavior in real-life scenarios.
These clues collectively suggest that the image is AI-generated. **</reason>**

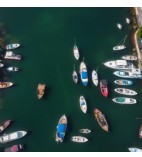

Please evaluate whether this image is an AI creation or something real, and provide an explanation.

[GT: Fake]

**<detect>**This is a fake image. **</detect>**
**<caption>**Image Description: The image depicts an aerial view of a marina with numerous boats docked in calm, greenish water. The boats vary in size and color, including white, blue, red, and brown. Some boats are moored close to the shore, while others are scattered across the water. The shoreline is visible on the right side of the image, with a paved walkway and some greenery. The water is relatively still, with no visible waves or ripples. **</caption>**
**<reason>**Based on the provided fake image, here are the reasons why this image is fake:
1. Line segments: The shadows cast by the boats do not consistently align with a single light source. Some boats have shadows that suggest the light is coming from the top left, while others have shadows indicating light from the top right. This inconsistency in lighting suggests that the image may have been artificially generated.
2. Edges: Upon closer inspection, some of the boat edges appear slightly blurred or pixelated, which is not typical for a high-resolution aerial photograph. This blurriness is particularly noticeable around the smaller boats and those closer to the shore.
3. Texture: The water texture appears overly smooth and uniform, lacking the natural variations and reflections that would be expected in a real aerial photograph. Additionally, the greenish hue of the water is unusually consistent, which is not typical for natural water bodies.
4. Distortion: Some of the boats, particularly those in the center of the image, exhibit slight distortions in their shapes. The proportions of these boats do not perfectly match those of real boats, suggesting artificial generation.
5. Overall Hue: The overall color tone of the image is somewhat unnatural. The greenish tint of the water and the vibrancy of the boat colors are exaggerated, which is a common characteristic of AI-generated images.
6. Clarity: The image shows signs of unnatural pixel blurriness, especially around the edges of the boats and the shoreline. This lack of sharpness is not typical for a high-resolution aerial photograph.
7. Perspective: The perspective of the image seems slightly off. The boats closer to the shore appear disproportionately large compared to those further out, which is not consistent with natural perspective rules.
8. Shadows: The shadows cast by the boats do not align with the expected direction of the light source. Some shadows are too sharp or too soft, indicating artificial generation.
By considering these clues, it becomes evident that the image is likely AI-generated rather than a genuine photograph. **</reason>**

*Figure 8.* **AIGI Detection Results of our GenShield.**

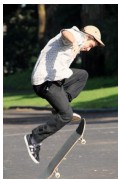

Please evaluate whether this image is an AI creation or something real, and provide an explanation.

[GT: Real]

**\<detect>**This is a real image. **\</detect>**
**\<caption>**Image Description: The image depicts a young man performing a skateboarding trick in an outdoor setting. He is captured mid-air, with his skateboard tilted at an angle beneath him. The man is wearing a light-colored cap, a short-sleeved plaid shirt, dark pants, and sneakers. The background shows a grassy area and trees, suggesting a park or a similar open space. The lighting appears natural, with shadows cast by the sunlight. **\</caption>**
**\<reason>**Based on the provided real image, here are the reasons why this image is real:
1. Line segments: The overall style, color, and details of the image are consistent. The lighting and shadows are natural and align with a single light source, which is the sun. The shadows cast by the skateboarder and the skateboard are consistent with the direction of the sunlight. The parallel lines of the skateboard and the ground converge appropriately, adhering to the rules of perspective.
2. Edges: The edges of the skateboarder, the skateboard, and the background elements are clear and consistent. There are no signs of unnatural pixel distribution or discontinuous edges that might suggest AI generation.
3. Texture: The textures in the image, such as the fabric of the skateboarder's clothing, the surface of the skateboard, and the grass in the background, appear natural and detailed. There are no repetitive patterns or blurry textures that would indicate AI generation.
4. Distortion: The objects in the image, including the skateboarder and the skateboard, exhibit natural shapes and proportions. There is no unnatural distortion that would suggest AI manipulation.
5. Overall Hue: The overall tone of the image is balanced and natural. The colors are neither overly vibrant nor too dull, and there are no inconsistencies in the hue that would suggest AI generation.
6. Clarity: The image has a high resolution with no visible compression artifacts. The details are clear, and there is no unnatural pixel blurriness or jagged edges.
7. Perspective: The perspective and proportional relationships in the image are consistent. The depth of field changes reasonably, with the skateboarder in focus and the background slightly blurred, adhering to the laws of physics.
8. Shadows: The shadows in the image are consistent with the direction of the sunlight. The skateboarder and the skateboard cast shadows that align with the light source, indicating that the image is real.
9. Physical Laws: The image adheres to physical laws. The skateboarder's pose and the position of the skateboard are realistic and physically plausible.
10. Body Structure: The skateboarder's body structure, including the head, neck, body, arms, hands, legs, and feet, appears anatomically correct. There are no signs of anatomical errors that might suggest AI generation.
11. Common Sense: The image does not violate common sense. The skateboarder's attire, the setting, and the action being performed are all consistent with a real-world scenario.
These clues collectively suggest that the image is real and not AI-generated. **\</reason>**

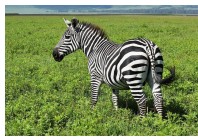

Please evaluate whether this image is an AI creation or something real, and provide an explanation.

[GT: Real]

**\<detect>**This is a real image. **\</detect>**
**\<caption>**Image Description: The image depicts a zebra standing in a lush, green field. The zebra is facing to the left, with its head slightly turned towards the camera. The field is covered in tall grass and scattered wildflowers, with a distant horizon visible in the background. The lighting suggests it is daytime, with natural sunlight illuminating the scene. **\</caption>**
**\<reason>**Based on the provided real image, here are the reasons why this image is real:
1. Line segments: The lighting and shadows in the image are consistent with a single light source, which is the sun. The zebra's stripes and the grass blades cast shadows that align with the direction of the sunlight. There are no unnatural distortions or inconsistencies in the line segments of the zebra's body or the surrounding environment.
2. Edges: The edges of the zebra and the grass are clear and well-defined. There are no signs of pixelation, discontinuous edges, or AI-generated artifacts. The transition between the zebra and the background is smooth and natural.
3. Texture: The texture of the zebra's fur and the grass is realistic and detailed. There are no repetitive patterns or blurry areas that would suggest AI generation. The grass appears natural and varied, with different shades of green and individual blades visible.
4. Distortion: The zebra's body and the surrounding environment do not exhibit any unnatural distortions. The zebra's proportions and the perspective of the field are consistent with real-world observations.
5. Overall Hue: The overall tone of the image is natural and consistent with outdoor lighting. The colors are neither overly vibrant nor too dull, and there are no inconsistencies in the hue that would suggest AI manipulation.
6. Clarity: The image has a high resolution with no visible compression artifacts. The details of the zebra's stripes and the grass are clear and sharp, with no signs of pixel blurriness or jagged edges.
7. Perspective: The perspective of the image is consistent and realistic. The zebra's size and position relative to the field and the background suggest a natural depth of field. The blurring between the foreground and background is gradual and conforms to physical laws.
8. Shadows: The shadows cast by the zebra and the grass are consistent with the direction of the sunlight. There are no unnatural or inconsistent shadows that would suggest AI generation.
9. Physical Laws: The image does not violate any physical laws. The zebra's stance and the behavior of the grass and shadows are consistent with real-world physics.
10. Common Sense: The image depicts a common and realistic scene of a zebra in its natural habitat. There are no elements that violate common sense or appear out of place.
These clues collectively suggest that the image is real and not AI-generated. **\</reason>**

*Figure 9.* **More AIGI Detection Results of our GenShield.**

| Abnormal Images | Initial Diagnose | Corrected Images | Termination Diagnosis |
|---|---|---|---|
| 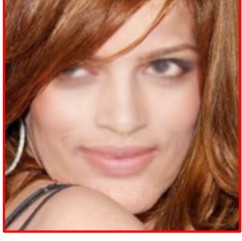 | <diagnosis> Upon examining the image. I have found: A woman with long brown hair and hoop earrings is depicted with her left eye appearing blurred at the junction with the edge of her bangs.\nTo elaborate, I have found the following artifacts. The left eye of the woman in the picture:The left eye of the woman in the picture appears blurred at the junction with the edge of the bangs. </diagnosis> <repair> | 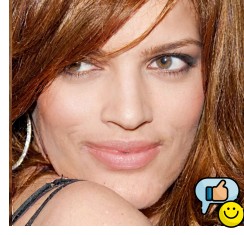 | <diagnosis> Upon examining the image, I have found: a close-up portrait of a woman with auburn hair and brown eyes, looking directly at the camera with a subtle smile. The skin texture, lighting, and hair strands appear entirely natural.\nTo elaborate, I don't notice any artifacts, this image is already a normal image. </diagnosis> <repair> |
| 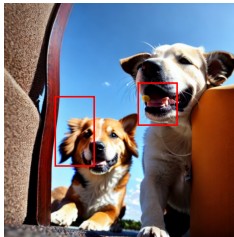 | <diagnosis> Upon examining the image. I have found: Two playful dogs are captured in the image, with the left dog's ear being asymmetrically shaped and the right dog holding a strange yellow object in its mouth.\nTo elaborate, I have found the following artifacts. Yellow puppy ears on left side:The left yellow puppy's left ear was abnormally shaped and asymmetrically sized from his right ear.Dogs on the right side of the picture:The dog on the right side of the picture has a strange yellow object in his mouth that shouldn't be there. </diagnosis> <repair> | 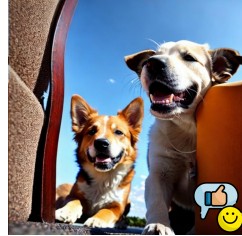 | <diagnosis> Upon examining the image, I have found: two happy dogs looking down at the camera against a clear blue sky. On the left is a fluffy, brown and white dog with pointed ears and a friendly expression. To the right is a larger, light-colored dog with its mouth open in a joyful pant. The low-angle perspective creates a playful and engaging scene.\nTo elaborate, I don't notice any artifacts, this image is already a normal image. </diagnosis> <repair> |
| 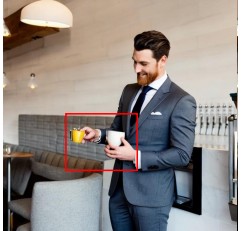 | <diagnosis> Upon examining the image. I have found: A man in a suit stands in a modern cafe holding a yellow cup in his left hand and a white cup in his right hand, with both hands showing fused and deformed fingers.\nTo elaborate, I have found the following artifacts. Boys left hand:The boy's left hand fingers were fused with the yellow cup and deformed, resulting in deformity.Boys right hand:The boy's right hand fingers were fused with the cup and deformed, resulting in deformity. </diagnosis> <repair> | 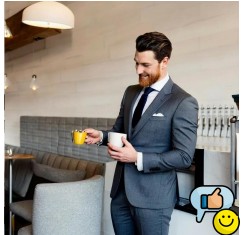 | <diagnosis> Upon examining the image, I have found: a well-dressed man in a grey suit and blue tie standing in a modern cafe. He is smiling while holding a white mug in one hand and a small yellow cup in the other.\nTo elaborate, I don't notice any artifacts, this image is already a normal image. </diagnosis> <repair> |
| 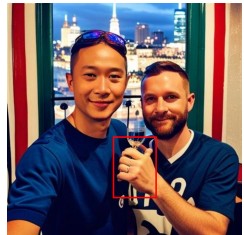 | <diagnosis> Upon examining the image. I have found: Two people pose for a photo in front of a cityscape backdrop, with one holding a wine glass that appears incomplete as if only half is visible.\nTo elaborate, I have found the following artifacts. Wine Glass:The edge of the wine glass disappeared and it looked like only half of it. </diagnosis> <repair> | 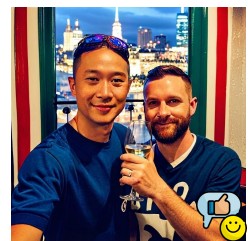 | <diagnosis> Upon examining the image. I have found: two men smiling for a selfie in front of a window with a city skyline at dusk. One man is holding a glass of white wine and wearing a wedding band.\nTo elaborate, I don't notice any artifacts, this image is already a normal image. </diagnosis> <repair> |
| 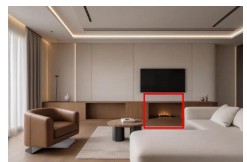 | <diagnosis> Upon examining the image. I have found: A modern living room features a sleek TV mounted on the wall above a cabinet, which unexpectedly contains a small fire, surrounded by minimalist furniture including a brown armchair, a white sofa, and a round coffee table.\nTo elaborate, I have found the following artifacts. Inside the cabinet underneath the TV:There should not be a fire in the cabinet below the TV stand. </diagnosis> <repair> | 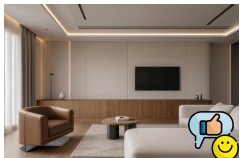 | <diagnosis> Upon examining the image. I have found: A modern living room features a sleek TV mounted on the wall above a cabinet, which unexpectedly contains a small fire, surrounded by minimalist furniture including a brown armchair, a white sofa, and a round coffee table.\nTo elaborate, I have found the following artifacts. Inside the cabinet underneath the TV:There should not be a fire in the cabinet below the TV stand. </diagnosis> <repair> |
| 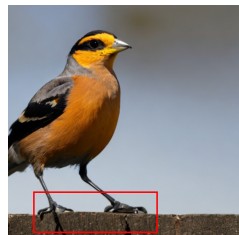 | <diagnosis> Upon examining the image. I have found: A vibrant bird perches on a wooden surface, with its right foot merging with the ground and the shadow of its left foot appearing naturally.\nTo elaborate, I have found the following artifacts. Bird's right foot:The bird's right foot merges with the ground.Shadow of the bird's left foot:The shadow of the bird's left foot doesn't extend abnormally out of reality. </diagnosis> <repair> | 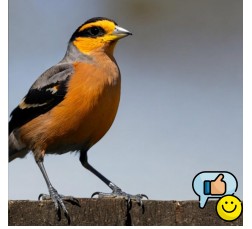 | <diagnosis> Upon examining the image, I have found a detailed photograph of a Bullock's Oriole perched on a wooden fence. The bird displays its characteristic vibrant orange plumage on its chest and face, contrasted by a black crown, eye line, and wings with white patches. Its greyish-blue legs and sharp beak are clearly defined against a soft, out-of-focus background.\nTo elaborate, I don't notice any artifacts, this image is already a normal image. </diagnosis> <repair> |
| 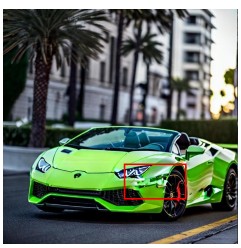 | <diagnosis> Upon examining the image. I have found: A vibrant green sports car is parked on a city street, featuring a color smudge under the right headlight causing distortion, a deformed right side exhaust grille differing from the left, and a right front cover lacking exhaust holes.\nTo elaborate, I have found the following artifacts. Right side of the car headlight:There is a color smudge under the right headlight of the car, causing distortion.Right side car exhaust grille:The style of the exhaust grille of the car on the right is obviously different from that on the left, and it looks deformed.Car right front cover:Compared with the left front cover, the right front cover of the car lacks exhaust holes, making it deformed. </diagnosis> <repair> | 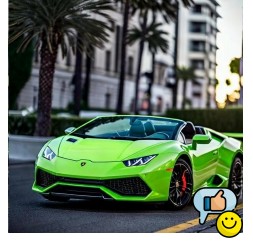 | <diagnosis> Upon examining the image, I have found a bright lime green Lamborghini Huracán Spyder parked on a street lined with palm trees and modern buildings. The lighting is soft, highlighting the car's sleek angles and black wheels.\nTo elaborate, I don't notice any artifacts, this image is already a normal image. </diagnosis> <repair> |

*Figure 10.* **Artifact Correction Dataset Samples.**

