# OpenReview forum: "GenShield: Unified Detection and Artifact Correction for AI-Generated Images"
_ICML.cc/2026/Conference — ICML 2026 regular_

### Official Review · Reviewer_ZbT9 · 2026-03-10

**Soundness:** 2
**Presentation:** 3
**Significance:** 3
**Originality:** 2
**Overall Recommendation:** 4
**Confidence:** 4

**Summary:**

This paper introduces GenShield, an autoregressive framework based on the BAGEL architecture that integrates AI-generated image (AIGI) detection and artifact correction. By introducing a curated dataset, GenShield-Set, and a Visual Chain-of-Thought (VCoT) curriculum learning strategy, the authors propose a closed-loop system that attempts to shift forensics from passive verification to active restoration.

**Compliance With Llm Reviewing Policy:**

Affirmed.

**Final Justification:**

I am overall OK with the current status of the paper and the discussions, and the detection and correction idea is attractive, so I keep my initial rating.

About the claim that "our method detects AI-generated images even without visible artifacts. As shown in Fig. 3(c) and Fig. 8, it correctly identifies images lacking clear semantic errors by analyzing low-level cues such as geometry, lighting, texture, edge inconsistencies, and local distortions.", yes the method may be good at detecting those AI images without visible artifacts, but my point is that the "deal with" these images is not perfect as the output "reasons" for these fakes are very likely not comprehensible by humans, which compromised the value and necessity of these "reasoning outputs".

**Key Questions For Authors:**

1. Why is a unified BAGEL architecture necessary? Is it possible to provide a head-to-head performance comparison against a strong, decoupled modular pipeline?
2. What is the impact of detection errors on correction quality? E.g., the error propagation from the detection stage to the restoration stage.
3. Is the model limited to the forensics of human-visible artifacts in AI generated image? From examples in the paper, it seems so. And how to tackle the other part of high-quality and not obvious fakes?

**Limitations:**

It seems to be a limitation that the method can only deal with human-visible artifacts.

**Strengths And Weaknesses:**

**Strengths**
The paper proposes an interesting "detect-and-repair" paradigm that successfully pivots digital forensics from passive classification toward active, restorative interventions. The introduction of GenShield-Set provides a high-quality, fine-grained benchmark that effectively addresses the data scarcity in AIGI artifact correction, and the framework’s closed-loop design represents a meaningful step toward integrating multimodal priors for unified forensic tasks.

**Weaknesses**
 While the integration of these two tasks is conceptually interesting, the paper has some concerns regarding the necessity of the unified architecture, the depth of its evaluation, and the clarity of its technical contributions.
1. Unjustified architectural choice: The core motivation for adopting a unified BAGEL architecture remains to be justified. It should demonstrate why this monolithic approach is superior to a modular "detector-plus-corrector" pipeline. A strong baseline comparison is essential here to justify the added complexity.
2. Marginal effectiveness of VCoT: The VCoT strategy, despite its added complexity, yields only marginal performance improvements in ablation studies. It is not adequately addressed whether this complexity is justified by the gains, or if it introduces unnecessary overhead.
3. Lack of empirical verification: The framework assumes an autoregressive dependency between detection and correction, yet the paper lacks empirical analysis of how detection errors (e.g., false negatives) propagate through the system. Specifically, the paper does not quantify how the correction performance degrades when the detector fails. Without a systematic analysis of these failure modes, it remains unclear whether the model is robust or is it sensitive to the inaccuracies of the initial detection stage.
4. Lack of theoretical depth: The technical description is abstract and lacks sufficient theoretical underpinning. While the method section outlines high-level optimization objectives, it does not provide a formal algorithmic pipeline that rigorously integrates detection and restoration. The underlying mechanics of the "closed-loop" interaction remain under-defined, making it difficult to assess the theoretical soundness of the proposed framework.

---

> ### Author Rebuttal · Authors · 2026-03-30
>
> Thank you for your valuable comments! If there are any additional comments to be added, please continue the discussion with us.
> >**W1 & Q1: Unjustified architectural choice**
> - **Why MoT instead of modular pipelines.** Modular pipelines separate understanding from an external editor (e.g., MetaQuery, BLIP3-o), requiring iterative passing of prompts, masks, or queries across modules. This process can discard fine-grained artifact cues and propagate diagnostic errors into subsequent edits. In contrast, BAGEL's architecture unifies detection and generation through shared self-attention and a common multimodal state, enabling artifact evidence to directly guide correction while allowing generative priors to support detection. The LEGION△ baseline in Tab. 2 exemplifies a detector-corrector pipeline, and our results show substantially better artifact correction performance.
> - **Alignment with VCoT closed-loop reasoning.** GenShield is not a single-step editor but follows a VCoT closed loop of "diagnose → correct → re-diagnose → (...) → stop". This requires alternating textual diagnosis and visual correction within a single sequence and learning an explicit stopping criterion. A unified sequential model is inherently better suited to this behavior than modular pipelines.
> - **Practical and empirical support.** BAGEL is a natural choice due to its pretraining on large-scale editing-with-thinking data, which aligns well with VCoT-style interleaved reasoning. Recent works (e.g., ThinkMorph, Uni-CoT) also adopt BAGEL-like architectures, further supporting its effectiveness. Because these modular pipelines (e.g., BLIP3-o) are not pretrained on interleaved data, they struggle to converge on VCoT-style tasks, leading to degraded performance and making direct comparison less meaningful. We will explore this issue in future work
>
> >**W2: Marginal effectiveness of VCoT**
> - To further demonstrate the effectiveness of VCoT, we conduct additional experiments. We adopt broader metrics, including image quality (Q-Insight), aesthetics (Aesthetic), and human preference (ImageReward, HPSv2.1). The results, shown below clearly demonstrate the significant gains brought by VCoT.
>
> ||Q-Insight ↑|Aesthetic ↑|ImageReward ↑|HPSv2.1 ↑|
> |-|-|-|-|-|
> |w/o VCoT|3.51|5.21|0.08|20.63|
> |Ours|3.89|6.34|0.23|22.15|
> >**W3 & Q2: Lack of empirical verification**
> - Because the reviewer's use of "detection" may refer either to the standalone detection task or to the diagnostic stage of correction, we address both cases.
> - **Between detection and correction tasks:** No error propagation occurs because the two are independent. They use different datasets and objectives and are executed separately at inference.
> - **Within the correction task (diagnosis → correction):** We do not observe significant error propagation for three reasons:
>   - During training, we place greater weight on image generation loss (`CE:MSE=0.25:1`), so correction is driven mainly by visual signals, with text serving only as auxiliary guidance.
>   - During inference, correction is conditioned on both image and diagnostic text tokens, but image tokens are more numerous, and we adopt a stronger image guidance scale than text guidance. Thus, even inaccurate diagnostic text does not cause catastrophic correction errors.
>   - With the VCoT strategy, an error introduced in one turn can be detected and corrected in later turns, preventing accumulation across steps.
>
> >**W4: Lack of theoretical depth**
> - At inference, GenShield follows an iterative diagnose–then–correct loop:
> ```
> Input: anomalous image I, prompt Q
> while true:
>   T_diag ← DetectExpert(I)
>   if T_diag indicates no artifact: break
>   I ← CorrectExpert(I, T_diag)
> return I
> ```
> - Detection expert produces diagnostic cues $\mathbf{T}_{\text{diag}}$ to guide correction, and each refined image is used for the subsequent diagnose, forming a closed loop.
> - During training, detection and correction are tightly coupled:
>   - **Shared representation**: Both experts operate within a unified transformer with shared self-attention, enabling direct interaction between diagnostic and image tokens.
>   - **Joint optimization**: They are jointly optimized via: $\mathcal{L}=\mathcal{L}\_{\text{correct}}+\lambda\mathcal{L}\_{\text{detect}}$. Detection provides structured guidance for correction, while correction enforces a real-image prior that improves detection sensitivity.
>
> > **Q3: Is the Model Limited to Visible Artifacts?**
> - No. Our method detects AI-generated images even without visible artifacts. As shown in Fig. 3(c) and Fig. 8, it correctly identifies images lacking clear semantic errors by analyzing low-level cues such as geometry, lighting, texture, edge inconsistencies, and local distortions.
> - As shown in Tab. 1, our method achieves an average detection accuracy of 98.8%, and obtains 97.9% and 96.7% on advanced generators FLUX and SD3.5, respectively, which typically produce images without obvious visible artifacts.

---

> > ### Author Rebuttal · Reviewer_ZbT9 · 2026-04-02
> >
> > Thank you for the detailed rebuttal. The responses address some of my concerns, and I appreciate the clarifications provided.
> > Regarding the “Unjustified architectural choice” point, the current explanation is somewhat unclear. While VCoT and interleaved dialogue differ in formulation, they are essentially both multi-turn interaction processes, which does not by itself justify the necessity of the BAGEL architecture over alternative designs. In addition, since LEGION differs in training data, it may not serve as a fully controlled baseline for this comparison. Finally, the theoretical depth is still in short.
> > Overall, I am satisfied with the clarifications and maintain my current score, as it appropriately reflects the paper’s contribution.

---

> > > ### Author Response · Authors · 2026-04-02
> > >
> > > Thank you for your prompt response, your recognition of our rebuttal, and your positive rating.  We provide the following clarifications in the hope of fully addressing your remaining concerns.
> > >
> > > > **Why choose BAGEL?**
> > >
> > > - Our choice of BAGEL is primarily motivated by its pretraining data and strong unified understanding–generation capability, rather than its architecture itself.
> > >
> > >   - **Pretraining data.** During pretraining, BAGEL is exposed to large-scale interleaved multimodal sequences involving dialogue, generation, and editing. This makes it a strong foundation for multi-turn interleaved reasoning, which aligns well with our VCoT setting. Its capability in interleaved text–image modeling has also been widely validated in downstream tasks, and many prior works (e.g., ThinkMorph, UniCoT) build upon this foundation.
> > >
> > >   - **Strong performance.** Compared to other unified models, BAGEL achieves near state-of-the-art performance in both generation and understanding.
> > >
> > > - **Scope of our contribution.**
> > >   Our goal is not to compare different unified architectures, but to explore the mutual reinforcement between detection and correction, the design of VCoT, and the construction of the dataset. Our method is not tied to a specific backbone; it is a general framework that can be extended to other strong backbones, including detector–corrector structures such as BLIP-3o.
> > >
> > >
> > > > **Comparison with LEGION.**
> > >
> > > - To further address your concerns, we fine-tuned LEGION (SDXL) using the GenShield-set. LEGION is used to produce anomaly diagnosis results, and SDXL then repairs the abnormal images based on these diagnoses. We made our best effort to fully align the training settings of the LEGION (SDXL) and GenShield and evaluated the correction performance using a wide range of metrics. The results are shown in the table below:
> > >
> > > - |               | HPSv3 ↑| CLIP-Score ↑| PickScore ↑| Q-Insight ↑| Aesthetic ↑| ImageReward ↑| HPSv2.1 ↑|
> > >   | ------------- | ----- | ---------- | --------- | --------- | --------- | ----------- | ------- |
> > >   | LEGION (SDXL) | 5.27  | 21.82      | 18.78     | 3.18      | 5.56      | -0.13       | 21.85   |
> > >   | Ours          | 6.20  | 22.12      | 18.86     | 3.89      | 6.34      | 0.23        | 22.15   |
> > >
> > > - The results demonstrate that, even under a unified evaluation protocol, our model significantly outperforms the diagnose+correct pipeline composed of LEGION (SDXL).
> > >
> > >
> > > > **More theoretical depth.**
> > >
> > > - We agree that formal theoretical analysis is an important direction, and we clarify that our work already provides both principled motivation and empirical evidence supporting the design.
> > >
> > >   - **Principled motivation.** Our method is not purely heuristic, but is grounded in a principled joint modeling view: detection provides structured diagnostic signals that guide correction, while correction introduces generative priors that align features with real-image distributions. Together, this forms a mutually reinforcing learning process, which can be interpreted as a form of multi-task representation regularization.
> > >
> > >   - **Architectural perspective.** From an architectural perspective, the shared self-attention mechanism enables bidirectional information flow between detection and correction experts, which is analogous to co-training or shared representation learning, a well-established paradigm for improving generalization.
> > >
> > >   - **Empirical support.** Importantly, our ablation results (Table 3) provide strong empirical support: training detection or correction alone leads to suboptimal performance, while joint training consistently improves both tasks. This validates our core hypothesis that the two tasks are mutually beneficial rather than independent.
> > >
> > >   - **Scope of theoretical analysis.** We note that deriving formal guarantees (e.g., convergence or optimality) for large-scale multimodal autoregressive models remains an open challenge in the community, and most prior works in unified understanding-generation frameworks also rely on empirical validation. Nevertheless, our formulation is fully differentiable and well-defined, based on standard autoregressive likelihood and flow-matching objectives.
> > >
> > > - We will include additional discussion in the revision to further formalize the connection between joint learning and representation alignment.
> > >
> > > We sincerely thank you again for your valuable time and look forward to further discussion.

---

### Official Review · Reviewer_UT1Y · 2026-03-10

**Soundness:** 3
**Presentation:** 3
**Significance:** 4
**Originality:** 4
**Overall Recommendation:** 5
**Confidence:** 4

**Summary:**

This paper propose a unified autoregressive framework that connects AI image detection and artifact correction, named GenShield. By coupling semantic understanding with pixel-level reconstruction, our method forms an end-to-end loop from artifact diagnosis to authenticity restoration, showing the synergistic effects between detection and correction, and constructs a specialized, high-quality dataset `GenShield-Set tailored for unified AIGI detection and correction.

**Compliance With Llm Reviewing Policy:**

Affirmed.

**Final Justification:**

The authors jointly study two previously separate tasks, AIGI detection and artifact correction, and propose a VCoT mechanism to enhance correction performance, along with a high-quality artifact correction dataset. Overall, I find this work interesting, and insightful. After rebuttal, all of my concerns have been addressed, so I am willing to raise my score from 4 (weak accept) to 5 (accept), and I am inclined to recommend acceptance.

**Key Questions For Authors:**

See Weakness & Question.

**Limitations:**

Yes， see supplementary material sec. A

**Strengths And Weaknesses:**

Strengths:
I think the author’s move to unify generation and understanding in AIGI forensics is highly valuable. It’s a promising approach and honestly, it’s right in line with my own thinking. However, I’m still a bit unclear on a few points.

Weaknesses & Questions :
1. The GenShield-set has two versions： Correct and Detection, but I'm curious if the two-dimensional data is cross-functional. For instance, the `partial correct' subset appears to have untapped potential for fine-grained localization in explainable forensics.  However, the paper just keeps them as two tasks linked by the framework.

2. In the construction pipeline of GenShield-Set-Correct, some annotations for 'full-correct' samples contain statements such as: 'I don't notice any artifacts; this image is already a normal image.' However, at the pixel level, these 'full-correct' samples are technically AI-generated images. Given the influence of the unified framework, could these annotations cause the Detection Expert to overlook pixel-level artifacts and instead over-fit to global semantics?

Although the current annotations, unified framework, and training strategy make it difficult to definitively determine if there is a tangible gain for the original forensics task, I believe this remains a worthwhile endeavor.

---

> ### Author Rebuttal · Authors · 2026-03-30
>
> Thanks for your valuable comments. We have tried our best to address the concerns you raised. If you have any further questions, please continue the discussion with us.
>
> > **Weakness & Question #1: Limited Cross-Functional Utilization of Correct–Detection Annotations**
>
> - The partial correct subset indeed has potential to serve as data for explainable forensics, as it essentially consists of AI-generated images containing semantic errors. However, this subset currently lacks explicit explanatory annotations required for explainable detection, which would incur additional annotation costs and may introduce further concerns regarding annotation quality. We thank the reviewer for this valuable suggestion and will explore this direction in future work.
>
> - In addition, our detection experiments strictly follow existing forensic benchmarks. Both the training and testing data are fully public, without involving any private datasets, ensuring reproducibility and fair comparison.
>
> > **Weakness & Question #2: Potential Semantic Bias from "Full-Correct" Annotations Affecting Detection Sensitivity**
>
> - **Task-level separation and design.** During both training and inference, artifact correction and AIGI detection are treated as relatively independent tasks with different input/output formats, optimization objectives, and prompts. Moreover, in our instruction design, correction reasoning is enclosed within `<diagnosis></diagnosis>`, while detection reasoning is enclosed within `<reason></reason>`. This explicit instruction-level separation helps mitigate the potential semantic bias raised by the reviewer.
>
> - **Training dynamics and supervision balance.** We provide detailed implementation in Appendix E and Tab. 6. From the training schedule, in Stage 2, the sampling weight for detection is increased to 2.0, while the terminate-state correction (i.e., "already normal") only accounts for 0.1. Therefore, such supervision is not dominant during training. In addition, the AIGI detection task remains active throughout both stages.
>
> - **Experimental evidence (quantitative + qualitative).** During experiments, we do not observe any degradation in detection performance.
>   - As shown in Tab. 1, our method achieves an average detection accuracy of 98.8%, and obtains 97.9% and 96.7% on advanced generators FLUX and SD3.5, respectively, which typically produce images without obvious visible artifacts.
>   - Fig. 3(c) and Fig. 8 further provide qualitative detection examples. Even when the input images do not exhibit visible artifacts, the model's explanations still capture low-level forensic cues, including geometry, lighting, texture, edge artifacts, and local distortions, rather than relying solely on high-level semantics.
>
>
> - **Additional Ablation study on potential bias.** To further address this concern, we conduct an additional experiment by removing the $\mathbf{T}_{\text{stop}}$ data in Stage 2 and evaluating detection performance. As shown in the table below, removing such supervision does not lead to any significant change in detection performance, suggesting that the potential bias is negligible.
>
>   - |                                 | Acc. | A.P. |
>   | ------------------------------- | ---- | ---- |
>   | w/o  $\mathbf{T}_{\text{stop}}$ | 98.7 | 99.8 |
>   | Ours                            | 98.8 | 99.8 |

---

> > ### Author Rebuttal · Reviewer_UT1Y · 2026-04-01
> >
> > My concerns have been adequately addressed. I am satisfied with the rebuttal and will increase my score accordingly.

---

> > > ### Author Response · Authors · 2026-04-02
> > >
> > > Thank you for your valuable feedback and your prompt response. We sincerely appreciate your recognition and your willingness to increase the score.

---

### Official Review · Reviewer_4p6j · 2026-03-10

**Soundness:** 3
**Presentation:** 3
**Significance:** 3
**Originality:** 3
**Overall Recommendation:** 4
**Confidence:** 4

**Summary:**

This paper introduces a unified framework (GenShield) for autoregressive visual artifact detection and correction, showing mutual benefit of the two tasks in a joint model. The model is based on a framework similar to BAGEL with dual tower and attention sharing, and proposes a VCoT based curriculum learning strategy to correct artifacts by first performing the diagnosis. The paper also proposes a visual artifact correction dataset and evaluation pipeline. Experiments show that the proposed method shows results outperforming state of the art and strong generalization performance.

**Compliance With Llm Reviewing Policy:**

Affirmed.

**Final Justification:**

Rebuttal addressed my concerns so I will keep my original positive rating.

**Key Questions For Authors:**

1. Is the proposed method able to localize the artifact regions? For example, through attention map visualization, etc.
2. It would be great if authors can suggest ways to improve the method further (i.e. how to fix remaining artifacts in some of the cases to make results even better?)
3. What would happen if the input image is artifact-free already, would the method makes any changes?

**Limitations:**

Yes. In Appendix.

**Strengths And Weaknesses:**

Strengths:
1. The paper tackles an interesting GenAI artifact detection/correction problem with a novel MLLM solution.
2. New dataset is proposed which will be helpful for the research community.
3. Experimental results are comprehensive and show clear advantages over previous methods. Especially, the Mean Acc and AP in Tab.1 and visual evaluation metrics in Tab.2 show clear improvements over existing methods.

Weaknesses:
1. The architecture seems largely following BAGEL. It would be great if authors can clarify difference over BAGEL in terms of architecture and training details.
2. In Fig.3, some of the results still have left-over artifacts even after the correction is done. (man drinking beer example has extra finger behind glass, woman with glasses where glass frame/leg is blurry and missing).
3. Missing reference: Lingzhi Zhang et al. "Perceptual Artifact Localization for Image Synthesis Tasks" ICCV 2023.

---

> ### Author Rebuttal · Authors · 2026-03-30
>
> Thanks for your valuable comments. We have tried our best to address the concerns you raised. If you have any further questions, please continue the discussion with us.
> >**W1: Clarifying Differences from BAGEL**
> - **Architecture.** We acknowledge that our model builds upon the MoT-style architecture introduced in BAGEL. Our contribution does not lie in redesigning the backbone, but in extending it into a unified framework that jointly models AIGI detection and artifact correction with shared representations and specialized experts.
> - **Training strategy.** We provide detailed training configurations in Appendix E and Tab. 6. Our key difference lies in the training paradigm. We adopt a two-stage curriculum learning with VCoT-based self-correction, which is fundamentally different from BAGEL. Specifically, Stage 1 jointly trains detection and instruction-guided correction, while Stage 2 introduces multi-step VCoT correction trajectories (initial/intermediate/terminate) and increases detection sampling. This enables the model to perform progressive correction, learn explicit stopping behavior, and jointly model detection and correction, which is not considered in BAGEL.
>
> >**W2 & Q2: Residual Artifacts Remain After Correction and Further Improvement**
> - We thank the reviewer for the observation. As noted in our Limitations, correction is still constrained by the current ceiling of SOTA image editing models (e.g., Nano Banana Pro), so very fine-grained details may remain imperfect. We will clarify this in the final version.
> - The remaining artifacts are mostly minor post-editing imperfections. In contrast, the major structural and logical errors have been corrected, including finger count and structure, left-right hand consistency, and the geometry of the glasses and eyes. Even strong commercial models (e.g., Nano Banana, GPT-Image, and Seedream) often fail to detect or fix such relatively noticeable artifacts. As an early exploration of artifact correction, we believe this performance is acceptable.
> - Additional qualitative examples in Figs. 6 and 7 further demonstrate the effectiveness of our method.
> - Future gains will likely come from better data quality and training strategies, including:
>   - **Stronger data generation**: use multiple advanced editors (e.g., GPT-Image, Nano Banana Pro, Seedream) per image and keep the best result after filtering.
>   - **Stricter data filtering**: remove samples with even minor artifacts.
>   - **Improved learning strategies**: incorporate reinforcement learning (e.g., Flow-GRPO) with suitable rewards to refine correction quality.
>   - **Quality-aware data curation**:  use score-based annotation to obtain samples of graded quality and adopt a multi-stage data strategy. Larger but noisier data are used for rapid convergence in early stages, while high-quality samples are progressively emphasized for later refinement.
>
> >**W3: Missing Relevant References**
> - Thank you for the suggestion. We will include and discuss this in the related work section of the final version.
>
> >**Q1: Does the method have the ability to localize artifacts?**
> - In fact, during the correction process, the model already provides artifact localization in textual form as part of its diagnostic reasoning. Detailed interleaved diagnosis–correction processes are illustrated in Figures 6 and 7 of the Appendix.
> - The diagnostic output consists of two components: (1) an image caption containing a coarse description of artifacts, and (2) fine-grained localization and explanation of individual artifacts. For example, in Fig. 6 (first row, Turn 1), the model first describes the overall scene ("a young man holding a book") and identifies an issue with distorted fingers. It then explicitly refers to the "boy's left hand" and "boy's right hand," providing detailed explanations of the specific problems in each region.
> - Furthermore, these localization cues can be extracted and fed into text-to-mask models (e.g., SAM 3) to obtain precise artifact segmentation masks.
>
> >**Q3: How Does the Method Behave on Artifact-Free Input Images?**
> - This scenario is already considered in our VCoT framework, which is precisely why we introduce the termination diagnosis mechanism. When an artifact-free image is input for correction, the model first performs a comprehensive analysis and concludes that the image is already normal and free of artifacts. It then outputs an image that is identical to the input, without making any modifications.
> - As illustrated in Figures 6 and 7, we provide detailed qualitative examples of the correction process. In the second-to-last turn, the model has already successfully removed all artifacts and obtained an artifact-free image. In the final turn, the model explicitly produces a termination diagnosis indicating that no artifacts remain, and outputs an unchanged image.

---

> > ### Author Rebuttal · Reviewer_4p6j · 2026-04-03
> >
> > The rebuttal addressed most of my concerns. I am still on the positive side and will keep my score.

---

> > > ### Author Response · Authors · 2026-04-07
> > >
> > > We are pleased to have addressed your concerns. Thank you for your recognition and for your prompt response.

---

### Official Review · Reviewer_xaPZ · 2026-03-10

**Soundness:** 2
**Presentation:** 2
**Significance:** 3
**Originality:** 2
**Overall Recommendation:** 3
**Confidence:** 4

**Summary:**

This paper attempts to unify AIGI forgery detection and controllable artifact correction within a closed loop framework. It further uses a Visual Chain of Thought mechanism with an explicit stopping criterion to enable step by step restoration, and provides a large-scale paired artifact restoration dataset together with a unified evaluation pipeline. The focus is not on proposing yet another standalone detector or restorer, but on establishing a mutually reinforcing mechanism and training paradigm between the two tasks.

**Compliance With Llm Reviewing Policy:**

Affirmed.

**Final Justification:**

I still have concerns regarding the authors’ claim that the issue with Table 1 is merely a misunderstanding. Moreover, I do not find the reproduced results in the rebuttal to closely match Table 1, given that many entries still differ by 5–13 points and some average rankings even change (e.g., NPR vs. LaRE, AIDE vs. RINE). Therefore, I maintain my WR rating.

**Key Questions For Authors:**

1. Could the authors clarify whether the detector results are truly retrained on Holmes-Set or directly taken from AIGI-Holmes Table 2, and explain why the reported AIGI-Holmes performance is substantially lower than in the original paper under the claimed same-condition setting?

2. Could the authors clarify what concrete evidence supports the claimed mutual reinforcement between forgery detection and artifact restoration, beyond the current language-based discussion?

3. Other concerns are listed in the Weaknesses.

**Limitations:**

yes

**Strengths And Weaknesses:**

Strengths:

1. Unifying AI-generated image forgery detection and artifact correction within a single framework is an interesting idea, as it can provide interpretable diagnostic signals during detection and potentially translate these signals into controllable restoration objectives.

2. The proposed artifact restoration dataset represents a meaningful contribution to the advancement of research on AIGI artifact correction.

Weaknesses:

1. The paper claims that all baselines are retrained under the same conditions on Holmes-Set. However, the reported results for the non-LLM-based AI image detectors appear to be directly copied from Table 2 of AIGI-Holmes. In addition, the performance reported for AIGI-Holmes is substantially lower than what is presented in the original AIGI-Holmes paper. These inconsistencies raise serious doubts about the validity of the experimental protocol and, consequently, the effectiveness of the proposed method.

2. The authors repeatedly emphasize the importance of synergy between forgery detection and artifact restoration. However, the discussion is largely subjective and does not provide substantive evidence or rigorous analysis to demonstrate that the two tasks genuinely reinforce each other in practice.

3. Can the proposed detection method still correctly identify AI-generated images after artifact restoration has been applied?

4. Given the claimed scale of the proposed dataset, would relying on human experts be prohibitively expensive? What are the criteria or guidelines used for quality assessment?

5. The discussion of related work is insufficient. For example, Section 2.2 merely lists three prior works in a stacked, descriptive manner.

---

> ### Author Rebuttal · Authors · 2026-03-30
>
> Thanks for your valuable comments. We have addressed your concerns to the best of our ability and welcome any further discussion.
> >**W1 & Q1: Concerns on the Reliability of Tab. 1**
> - The non-LLM baselines in Tab. 1 are adopted from Tab. 2 of [1]. We follow that setup and use the official implementations to train AIGI-Holmes, FakeVLM, and Qwen2.5VL-7B. All methods in Tab. 1 are evaluated under the Tab. 2 protocol of [1].
> - To address the concern, we retrained all non-LLM baselines in Tab. 1 under the same protocol as our LLM-based methods, eliminating possible protocol mismatch. The reproduced results (accuracy) shown below closely match those in Tab. 1, confirming the consistency of evaluation and the robustness of our conclusions.
>
> |Method|Janus|Janus-Pro-1B|Janus-Pro-7B|Show-o|LlamaGen|Infinity|VAR|PixArt-XL|SD3.5-Large|FLUX|Mean|
> |-|-|-|-|-|-|-|-|-|-|-|-|
> |CNNSpot|77.4|67.7|87.2|80.7|65.7|91.0|61.5|78.3|61.0|81.9|75.2|
> |AntifakePrompt|74.1|89.9|91.9|84.1|94.1|90.8|89.8|84.4|91.3|61.8|85.2|
> |UnivFD|85.8|94.0|92.5|82.4|86.3|83.3|62.5|78.4|92.0|73.4|83.1|
> |NPR|64.6|66.8|81.3|93.9|94.2|97.2|88.8|98.0|94.5|93.2|87.2|
> |LaRE|65.9|73.6|97.7|76.8|92.3|71.2|94.5|79.3|90.1|79.1|82.1|
> |RINE|91.9|99.0|99.1|97.7|99.0|99.8|87.5|98.9|97.5|96.9|96.7|
> |AIDE|93.0|92.5|96.0|95.3|98.8|98.6|94.7|98.8|98.9|95.2|96.2|
> - **Regarding AIGI-Holmes:**
>   - Its original performance benefits from a **Collaborative Decoding** strategy, which ensembles LLM predictions with a CLIP-based expert (Eq. 6 in [1]). This introduces a multi-model ensemble, making direct comparison with single-model methods **less fair**.
>   - Therefore, we report results without ensembling, following the authors' released code (which does not include this strategy), and all results are fully reproducible. This protocol is also adopted in other works [2,3]. If similar ensembling were applied, our performance would further improve. We will clarify this explicitly in the final version.
>
> >**W2 & Q2: Lack Evidence for Detection–Correction Synergy**
> - Tab. 3 evaluates the synergy between AIGI detection and artifact correction. Compared with Only Det and Only Corr, our full model consistently improves both detection and correction.
> - To further demonstrate the synergy, we conduct additional experiments. For detection, we evaluate accuracy on unseen advanced closed-source generators from the AIGI-Now dataset. For correction, we adopt broader metrics, including image quality (Q-Insight), aesthetics (Aesthetic), and human preference (ImageReward, HPSv2.1).
> - The results shown below consistently confirm that combining detection and correction improves both tasks.
>
> ||GPT4o|JiMeng|Keling|MiniMax|Mean|
> |-|-|-|-|-|-|
> |Only Det.|85.4|82.9|88.1|94.6|87.8|
> |Ours|92.2|88.3|94.3|98.6|93.4|
>
> ||Q-Insight ↑|Aesthetic ↑|ImageReward ↑|HPSv2.1 ↑|
> |-|-|-|-|-|
> |Only Corr.|3.12|5.17|-0.37|19.61|
> |Ours|3.89|6.34|0.23|22.15|
> >**W3: Detection Robustness After Artifact Correction**
> - Artifact correction removes semantic errors but may retain unseen low-level artifacts, which remain detectable by forensic models.
> - The correction process follows an autoregressive generative paradigm (similar to Janus, Show-o, LlamaGen in Tab. 1), so corrected images remain within the detectable distribution.
> - We randomly sampled 500 corrected images and evaluated them using our detector, achieving 99.2% accuracy and confirming strong robustness after correction.
>
> >**W4: Annotation Cost & Quality Assurance**
> - **Human verification cost.** We incorporate human expert verification during GenShield-Set construction to ensure high-quality corrections. Such expert involvement is important for building a high-quality dataset. A team of 20 annotators completed filtering in 1 day, removing ~9% low-quality samples. This is an acceptable cost.
> - **Quality control protocol.** Appendix. D.1 describes the construction pipeline and quality criteria (e.g., artifact removal, consistency, structure). Each sample was reviewed by three experts and retained only when at least two judged it artifact-reduced and semantically consistent with the original image.
> - **Necessity of human experts vs. MLLMs.** To assess the necessity of human experts, we evaluated advanced MLLMs (GPT-5.2, Gemini 3.0, Qwen3VL-8B) as automated filters. The models were asked, under same guidelines, to classify human-verified 300 artifact-containing and 300 artifact-free images as "corrected" or "uncorrected." Their disagreement rates with human judgment were 53%, 38%, and 41%, respectively, indicating a substantial gap from human filtering. Notably, many samples misclassified as "uncorrected," leading to high rejection rates and fewer usable training samples.
>
> >**W5: Related Work**
> - We thank the reviewer and will expand the related work discussion for a more comprehensive coverage in the final version.
>
> **Reference:** [1] Zhou et al., AIGI-Holmes, 2025. [2] Lin et al., Seeing Before Reasoning, 2025. [3] Wang et al., TranX-Adapter, 2026.

---

> > ### Author Rebuttal · Reviewer_xaPZ · 2026-04-02
> >
> > Thank you for the authors’ response. Some of my concerns have been addressed. However, I still have questions about the credibility of the experimental results in Table 1.
> >
> > Specifically, on page 6 of the paper, around line 290, the paper explicitly states: *“To ensure a fair comparison, all methods were retrained on this dataset under identical conditions.”* However, in the rebuttal, the authors state: *“To address the concern, we retrained all non-LLM baselines in Tab. 1 …”*, and the reported results are different from those in the paper. This **inconsistency between the statements and the results** raises concerns about the reliability of the experimental setting and reporting.
> >
> > Moreover, the rebuttal also states: *“The non-LLM baselines in Tab. 1 are adopted from Tab. 2 of [1].”* This further **contradicts the claim in the paper** that all methods were retrained under identical conditions to ensure fairness.
> >
> > If I have misunderstood any part of the above, I would appreciate it if the authors could further clarify the experimental setup and the source of the reported results.

---

> > > ### Author Response · Authors · 2026-04-02
> > >
> > > Thank you for your response, and we are glad that some of your concerns have been addressed.
> > >
> > > Regarding the statement "all methods were retrained on this dataset under identical conditions" on line 290, page 6 of the paper, **there appears to be a misunderstanding.** What we intended to emphasize is that all compared methods are trained on Holmes-Set, rather than relying on their original pretrained weights. Since [1] has already retrained all non-LLM-based detectors in Table 2 on Holmes-Set, we reuse those results. In conclusion, in our Table 1 results, the retraining of non-LLM baselines was conducted by the authors of [1], while the retraining of LLM-based baselines was performed by us, as clarified in our rebuttal.
> > >
> > > In the manuscript, our intention was to emphasize the fairness of the comparison setting. We did not claim that all retraining was performed by us, nor did we state that "we retrained all baselines." **Therefore, the clarification provided in the rebuttal is not in conflict with the original text.**
> > >
> > > To further ensure the rigor of our evaluation, we additionally reproduced all non-LLM baselines in Table 1 during the rebuttal phase. The reproduced results closely match those in the original Table 1, indicating that our evaluation protocol is strictly aligned with Table 2 of [1], and is both accurate and fair.
> > >
> > > To avoid further confusion, **we will revise** the original statement
> > > "To ensure a fair comparison, all methods were retrained on this dataset under identical conditions."
> > > to:
> > > "For fair comparison, all methods were retrained on this dataset under identical conditions. Among them, all non-LLM-based detectors are sourced from [1], while we retrain all LLM-based detectors under the same training settings."
> > >
> > > We will also include our reproduced results in the Appendix. In addition, we will release all training code, logs, and checkpoints of both our GenShield and other comparison methods to ensure full reproducibility after the review process is completed.
> > >
> > > **We emphasize that all our results and experimental procedures are fair, sound, and reproducible**. We sincerely hope this clarification addresses your concerns, and we welcome further discussion.

---

### Decision · Program_Chairs · 2026-04-30

**Decision:**

Accept (regular)

**Comment:**

In this work, the authors propose a unified framework for explainable AI-generated image (AIGI) detection and controllable artifact correction, using a visual chain-of-thought-based (VCoT) curriculum learning strategy for multi-step diagnose-then-repair correction. Reviewers acknowledged the novelty of the proposed method (xaPZ, 4p6j, UT1Y, ZbT9), the artifact restoration dataset (xaPZ, 4p6j, ZbT9), and the comprehensive experimental results (4p6j). However, several concerns were raised, including limited references (xaPZ), issues with dataset annotation (UT1Y), an unclear explanation of the architectural choice to follow BAGEL (4p6j, ZbT9), an incorrect description of the experimental setup (xaPZ), insufficient experimental results, explanation, and analysis (xaPZ, ZbT9), and the marginal effectiveness of VCoT (ZbT9). In the rebuttal, the authors provided detailed explanations and additional experimental results that addressed most of these concerns. However, reviewer xaPZ still questioned the claim that all results in Table 1 were retrained, noting that some discrepancies remain despite the reproduced results in the rebuttal. The paper finally received one weak reject, two weak accepts, and one accept. Reviewer ZbT9 also remained concerned that the explanations generated by the proposed method for detecting AI-generated images without visible artifacts may be suboptimal and less comprehensible by human. AC agree with  xaPZ  that the authors have the responsibility to clearly describe the experimental setup without any ambiguities. Considering these factors, the paper is recommended for weak accept, and the authors are encouraged to further polish the paper to address the remaining concerns raised by reviewers xaPZ and ZbT9.